# Three-photon in vivo imaging of neurons and glia in the medial prefrontal cortex with sub-cellular resolution
Falko Fuhrmann[1,12], Felix C. Nebeling[1,2,12], Fabrizio Musacchio[1,12], Manuel Mittag[1], Stefanie Poll [1,3], Monika Müller[1], Eleonora Ambrad Giovannetti [1], Michael Maibach[1], Barbara Schaffran[4], Emily Burnside [4], Ivy Chi Wai Chan [5,11], Alex Simon Lagurin[5], Nicole Reichenbach[6], Sanjeev Kaushalya[7], Hans Fried[7], Stefan Linden[8], Gabor C. Petzold [6,9], Gaia Tavosanis [5,10,11], Frank Bradke [4] & Martin Fuhrmann [1] ✉

The medial prefrontal cortex (mPFC) is important for higher cognitive functions, including working memory, decision making, and emotional control. In vivo recordings of neuronal activity in the mPFC have been achieved via invasive electrical and optical approaches. Here we apply low invasive three-photon in vivo imaging in the mPFC of the mouse at unprecedented depth. Specifically, we measure neuronal and astrocytic $Ca^{2+}$-transient parameters in awake head-fixed mice up to a depth of 1700 μm. Furthermore, we longitudinally record dendritic spine density ($0.41 \pm 0.07$ μm$^{-1}$) deeper than 1 mm for a week. Using 1650 nm wavelength to excite red fluorescent microglia, we quantify their processes' motility ($58.9 \pm 2\%$ turnover rate) at previously unreachable depths (1100 μm). We establish three-photon imaging of the mPFC enabling neuronal and glial recordings with subcellular resolution that will pave the way for novel discoveries in this brain region.

The invention and application of two-photon (2P) laser scan microscopy for deep tissue in vivo imaging has greatly advanced our understanding of the brain[1,2]. It has been applied to visualize and analyze different cell types in various brain regions and model organisms[3–8]. Indicators, either synthetic or genetically encoded, have been applied to measure spatio-temporal dynamics of structure and function in networks of cells, individual cells or sub-cellular structures[9–22]. However, light scattering limited the depth penetration of two-photon excitation in brain tissue usually to less than a millimeter with cellular resolution[23,24]. Using higher wavelengths above 1300 nm for three-photon (3P) excitation allowed this limit to be exceeded in the brain[25]. Even functional $Ca^{2+}$-imaging of hippocampal CA1 neurons became possible in young mice[26]. In addition to neurons, non-neuronal cells, including astrocytes, oligodendrocytes and microglia have been imaged at increased depth in and below the somatosensory cortex[27,28]. Advantages of 3P-imaging have also been demonstrated in different model organisms, including zebrafish and *Drosophila*[29–32].

A brain region that has not been targeted with 3P-imaging up to now is the medial prefrontal cortex. The mPFC is involved in several higher cognitive functions, including memory, decision making and emotions[33–35]. Disturbances in the mPFC are hallmarks of different diseases including schizophrenia[36], autism spectrum disorders[37], and Alzheimer's disease[38]. Hence, this region is of high relevance for many neuroscientists. Optical access to record neuronal activity of mPFC neurons can be achieved by two invasive approaches. First, implantation of gradient refractive index (GRIN) lenses in combination with head-held microscopes can be used to record from mPFC neurons[39–41]. GRIN lenses are available with different diameters ranging from 500 μm up to more than a millimeter[42]. Cortical regions above the mPFC have to be penetrated and are partially damaged by their implantation. Second, microprisms have been implanted into the fissure between both hemispheres compressing the contralateral hemisphere opposite to the imaged ipsi-lateral hemisphere[43]. Both methods are

[1]Neuroimmunology and Imaging Group, German Center for Neurodegenerative Diseases (DZNE), Bonn, Germany. [2]Department of Neurooncology, Center for Neurology, University Hospital Bonn, Bonn, Germany. [3]IEECR, University Clinic Bonn, Bonn, Germany. [4]Axon Growth and Regeneration Group, German Center for Neurodegenerative Diseases (DZNE), Bonn, Germany. [5]Dynamics of Neuronal Circuits Group, German Center for Neurodegenerative Diseases (DZNE), Bonn, Germany. [6]Vascular Neurology Group, German Center for Neurodegenerative Diseases (DZNE), Bonn, Germany. [7]Core Research Facilities and Services, Light Microscope Facility, German Center for Neurodegenerative Diseases (DZNE), Bonn, Germany. [8]Department of Physics, Nanophotonics, University of Bonn, Bonn, Germany. [9]Division of Vascular Neurology, University Hospital Bonn, Bonn, Germany. [10]LIMES, University of Bonn, Bonn, Germany. [11]Department of Developmental Biology, RWTH Aachen University, Aachen, Germany. [12]These authors contributed equally: Falko Fuhrmann, Felix C. Nebeling, Fabrizio Musacchio. ✉e-mail: martin.fuhrmann@dzne.de

restricted by their limited imaging depth (approximately one hundred microns) starting from the implant. Limited imaging depth similarly applies for sub-cellular resolution in vivo imaging of dendritic spines in the PFC. Even with invasive microprism implantation only apical dendrites of L5 pyramidal neurons in the mPFC could be resolved with sub-cellular resolution[44,45].

Here we performed three-photon in vivo imaging through a cranial window to overcome these limitations. We achieved sub-cellular resolution imaging to measure spine density on the same L5 basal dendrites in the prelimbic area of the mPFC over a week. We recorded neuronal and astrocyte activity via $Ca^{2+}$-imaging throughout the entire dorsal column reaching the infralimbic areas of the mPFC. Time-lapse imaging of td-Tomato expressing microglia and their fine processes at 1650 nm excitation wavelength at a depth of >1000 μm, demonstrated the low invasiveness of our 3P-imaging approach, and revealed the kinetics of microglial fine processes' motility in the mPFC.

## Results

### Access of the medial prefrontal cortex with 3P-imaging at 1.6 mm depth

We built a 3P-microscope suitable for in vivo imaging as shown in Fig. 1a. The 3P-microscope consisted of a Thorlabs Bergamo multiphoton setup and a SPIRIT/NOPA laser combination from Spectra Physics tunable between 1300 and 1700 nm (Newport, SPIRIT 1030-70 and NOPA-VISIR + I). The laser repetition rate was maintained at 2 MHz and the microscope was equipped with 900–1900 nm coated optics in the primary scan path to enable transmission of 1300 to 1700 nm excitation light. To validate the capabilities of our 3P-microscope for in vivo imaging of the mPFC we chronically implanted rectangular cranial windows above the sagittal sinus to gain optical access to all areas and layers of the mPFC (Fig. 1b). Imaging was performed in head-fixed anesthetized YFP-H transgenic mice expressing yellow fluorescent protein (YFP) in a subset of excitatory neurons in different layers of the mPFC (Fig. 1c). These experimental conditions enabled us to image up to 1.6 mm deep from the brain surface, covering anterior cingulate, prelimbic and infralimbic areas (Fig. 1d; Supplementary Video 1). To compare 3P-imaging with 2P-imaging, a Ti:sapphire laser (Chameleon Ultra II, Coherent, Santa Clara, USA) was additionally integrated into the excitation light-path. The identical z-stack was imaged sequentially with 1300 nm excitation (3P) and 920 nm excitation (2P) (Fig. 1e; Supplementary Videos 2, 3). We achieved imaging two times deeper with 3P-excitation compared to 2P-excitation, using a maximum laser power of 200 mW at maximum depths. The necessary excitation power at 920 nm (2P) increased up to an imaging depth of 300 μm and started to plateau from 400 to 700 μm depth. The necessary excitation power at 1300 nm wavelength (3P) was low up to a depth of 400 μm, then exponentially increased up to a depth of 800 μm and started to plateau at 1000 μm (Fig. 1f). The signal to background ratio (SBR) of 2P-imaging rapidly decreased from 400 μm onwards to less than two at 500 to 600 μm depth. In contrast, the SBR of 3P-imaging only slightly decreased up to a depth of 1000 μm. Only at depths deeper than 1000 μm, the SBR started to decline (Fig. 1f). Plotting the ratio of SBR and laser power over imaging depth shows that 3P imaging is superior to 2P imaging throughout all imaging depths (Fig. 1g).

To achieve high penetration depth with high SBR, we optimized the microscope. Pulse broadening of femtosecond lasers, due to several optics included in the light path, significantly reduces excitation efficiency. Therefore, we included a prism-based compensator from APE (Femto-Control, APE) for dispersion compensation (Supplementary Fig. 1a). Dispersion compensation was adjusted under visual control of the interferogram measured after the objective. For 1300 nm, a negative chirp and for 1700 nm, a positive chirp was systematically adjusted according to the manufacturer's instructions. To probe its effect on advanced in vivo image quality, an identical z-stack was imaged subsequently with pulse compression ON and OFF (Supplementary Fig. 1a). The comparison of fluorescence histograms at 1 mm depth with or without pulse compression

showed more counts of high intensity fluorescence, when pulse compression was switched ON (Supplementary Fig. 1b). Signal to background ratio was improved by compensation for group delay dispersion throughout all imaging depths (Supplementary Fig. 1c).

Furthermore, we used a 25x water immersion microscope objective with a numerical aperture of 1.05 and transmission up to 1700 nm (XLPLN25XWMP2, Olympus) and measured its SBR in comparison to a custom Zeiss 20x, and Nikon 16x objective, in an identical subsequently recorded cortical volume (Supplementary Fig. 1d, e). The comparison of fluorescence histograms of the three objectives at 1 mm depths shows highest fluorescent intensities for the Olympus 25x (Supplementary Fig. 1d). Olympus 25x and Zeiss 20x show similar and enhanced SBR compared to Nikon 16x at all imaging depths (Supplementary Fig. 1e). Further improvement was achieved by increasing the size of detection optics (size of dichroic, collection lens, BP-filters), which increased the angle to collect more non-ballistic photons in the emission path (Supplementary Fig. 1f–i). To demonstrate the efficiency, an identical z-stack was recorded in the same mouse increasing the angle from 8° to 14° (Supplementary Fig. 1f). Highest fluorescence intensities were measured at 1 mm depths with the 14° detector port (Supplementary Fig. 1g). Highest SBR was achieved with the largest (14°) angle (Supplementary Fig. 1h, i).

Next, we attempted to access the hippocampus with 3P-imaging through the intact cortex (Supplementary Fig. 2). Since it is difficult to penetrate the highly myelinated corpus callosum in adult mice (4 months old), we chose a lateral imaging position to circumvent it (Supplementary Fig. 2a). This enabled us to image hippocampal CA1 neurons deeper than 1 mm. In contrast, with 2P-imaging, it was not possible to record YFP-expressing neurons deeper than 700–800 μm (Supplementary Fig. 2a–c). In younger mice (2.5 mo) 3P-imaging enabled imaging up to 1.3 mm with subcellular resolution of dendritic spines in dorsal CA1 (Supplementary Fig. 3, Supplementary Video 4).

Similar to imaging through the corpus callosum, the spinal cord represents an imaging challenge due to optical scattering from myelinated dorsal axons. Therefore, 2P-imaging approaches have been limited to superficial spinal grey matter laminae or ascending dorsal fiber tracts[46,47]. We tested our 3P-microscope in an adult (3 months old) mouse spinal cord. In a GFP-M mouse, following dorsal laminectomy and dura removal, a spinal cord window was implanted (Supplementary Fig. 4a). Vertebra were stabilized and the objective placed above grey matter dorsal horn (Supplementary Fig. 4b). Imaging depths of 390 μm were possible, including axons, cell bodies, dendrites and spines (Supplementary Fig. 4c, Supplementary Video 5). Thus, consistent with prior reports[48], it was not possible to achieve the depths of tissue penetration in the spinal cord that we observed in other CNS regions. However, our system allowed for a detailed view of neuronal architecture in the spinal cord grey matter. These data demonstrate access to the mPFC using 3P in vivo imaging up to a depth of 1600 μm with a chronically implanted cranial window as well as its limitations caused by light scattering myelinated axons.

### Time-lapse 3P-imaging of dendritic spines in the prelimbic area of the mPFC

Recording of subcellular structures, including dendritic spines in deep brain regions such as the prelimbic region of the mPFC can be achieved by implantation of microprisms[44,45]. However, the implantation of microprisms is an invasive procedure accompanied by tissue compression or lesions, if incisions or aspirations are necessary for the positioning of the implant. 3P-imaging promises to diminish the invasiveness for subcellular resolution imaging. Therefore, we recorded structural plasticity of dendritic spines in prelimbic areas of the mPFC. We implanted a rectangular cranial window in an aged (16 months old) thy1-GFP-M transgenic mouse (Fig. 2a) sparsely expressing green fluorescent protein (GFP) in a subset of excitatory neurons of the cortex (Fig. 2b). As in thy1-YFP-H mice, we reached mPFC prelimbic area in thy1-GFP-M transgenic mice with 3P-imaging (Fig. 2c, Supplementary Video 6). Dendritic spines on basal dendrites of layer 5 neurons were clearly visualized at a depth of 900–1100 μm (Fig. 2d). The measured spine density was 0.41 ± 0.07 spines per micrometer (Fig. 2d). Of

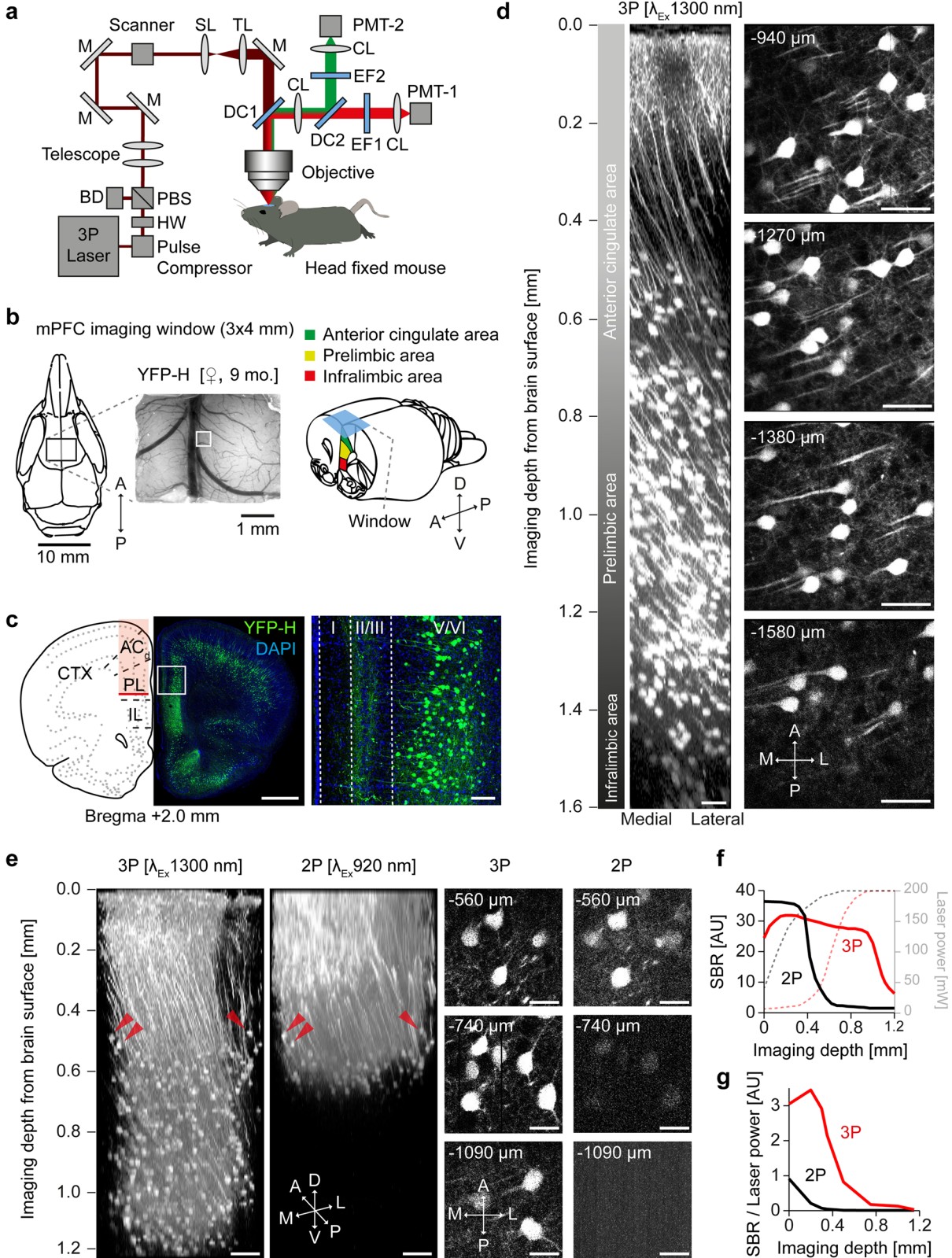

note, we did not use any adaptive optics to correct for wavefront distortions. Furthermore, we repetitively imaged the same dendritic spines over a week visualizing their structural plasticity at a depth of 1000 μm below the brain surface (Fig. 2e). These data show that 3P-imaging enables subcellular resolution imaging of structural plasticity of spines over a time period of one week in the prelimbic area of the mPFC.

**Recording microglial fine processes' motility in mPFC**

Microglia are innate immune cells in the brain. The discovery of the motility of their fine processes opened an entirely new research field almost 20 years ago[10,13]. Up to now, microglia have been imaged in various brain regions, however, the recording depth with 2P-imaging was limited to approximately 300–400 μm. To compare the recording depth between 2P- and

**Fig. 1 | Validation of three-photon imaging of mPFC in vivo. a** Setup scheme with light path and optical components. (BD: Beam dump CL: Collection Lens, DC: Dichroic, EF: Emission Filter, HW: Half wave plate, M: Mirror, PBS: Polarizing beam splitter, SL: Scan Lens, TL: Tube Lens). **b** mPFC imaging window position on the scull and representative image of the brain surface and vasculature (left, indicated ROI of z-stack shown in **d**); scheme of the anatomical localization of imaging window in relation to the mPFC subareas (right). **c** Confocal image of a coronal section 2 mm anterior of bregma and expression pattern from a YFP-H transgenic mouse in mPFC. Magnification and cellular YFP expression in layer V/VI of mPFC prelimbic area. Scale bars: 1 mm, 100 μm. **d** 3D reconstruction of 320 *x–y* frames from brain surface to 1600 μm below taken at a depth increment of 5 μm (left) and individual frames at stated imaging depths (right). Scale bars: 50 μm. **e** 3D reconstruction and side by side comparison of identical z-stacks recorded with 3P 1300 nm and 2P 920 nm excitation with individual frames at stated imaging depths. Scale bars: 100 μm (left panels), 15 μm (right panels). **f** Signal to background ratio (SBR) as a function of imaging depth for the z-stacks shown in e. Dashed lines display laser power as a function of imaging depth for the z-stacks shown in (**e**). **g** SBR/Laser Power as a function of imaging depth for the z-stacks shown in (**e**).

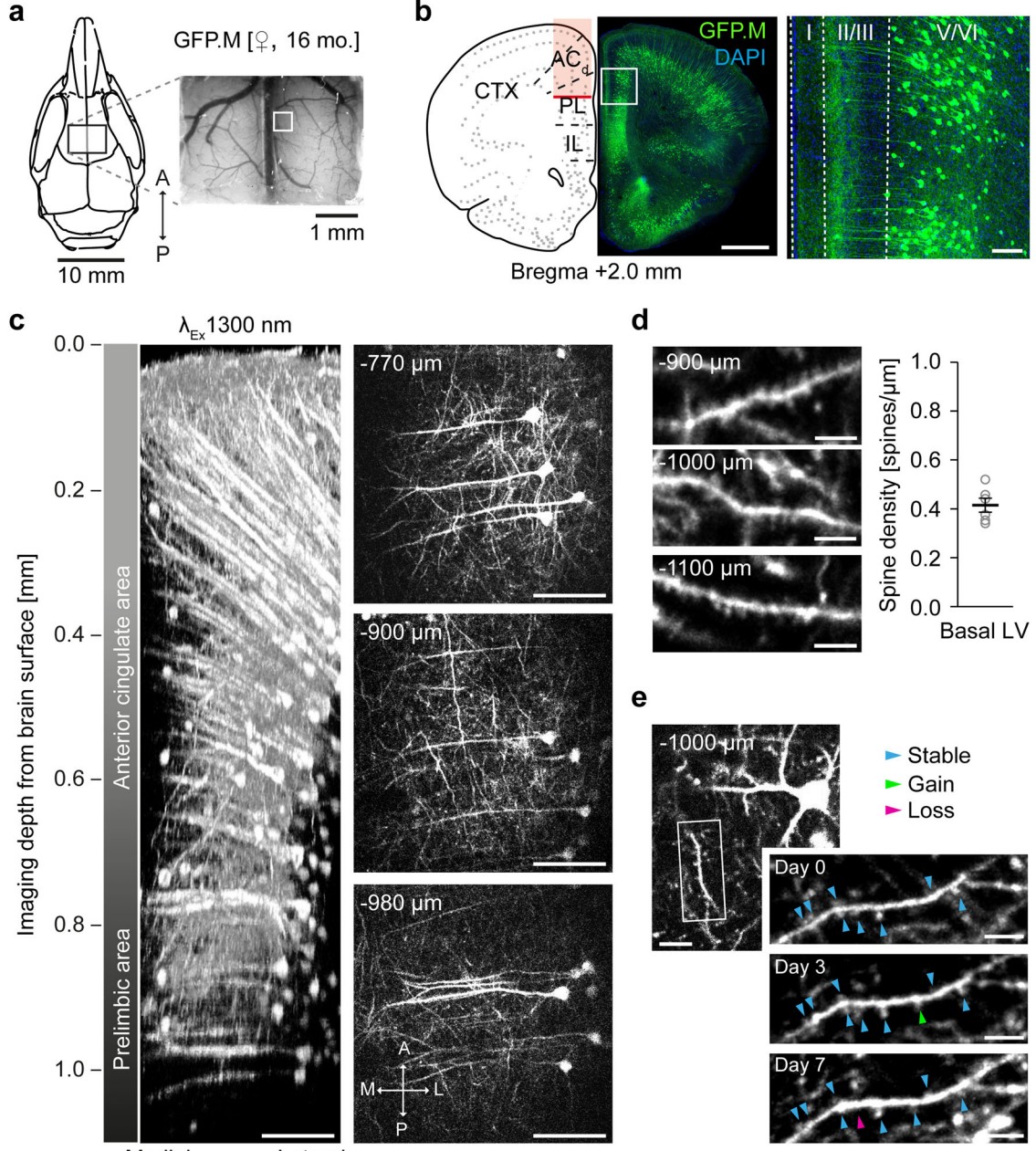

**Fig. 2 | Longitudinal three-photon imaging of dendritic spines in mPFC. a** mPFC window positioning and image of the brain surface and vasculature in a Thy1-GFP-M mouse. **b** Schematic and coronal section illustrating the imaging area (left). Zoom of the marked ROI in (**b**) showing cellular GFP expression in layer V/VI of mPFC prelimbic area (right). Scale bars: 1 mm, 100 μm. **c** 3D reconstruction and exemplary images of a z-stack acquired at the ROI marked in (**a**). Scale bars: 100 μm. **d** Exemplary images of three different basal dendrites of L5 neurons with spines in mPFC prelimbic area (left) and the average spine density (*n* = 6 dendrites, mean ± SEM, total length = 280.5 μm). Scale bars: 10 μm. **e** Longitudinal (7 d) imaging of dendritic spines on L5 neurons in mPFC prelimbic area. Arrowheads indicate stable (blue), gained (green) and lost (magenta) spines. Scale bars: 10 μm.

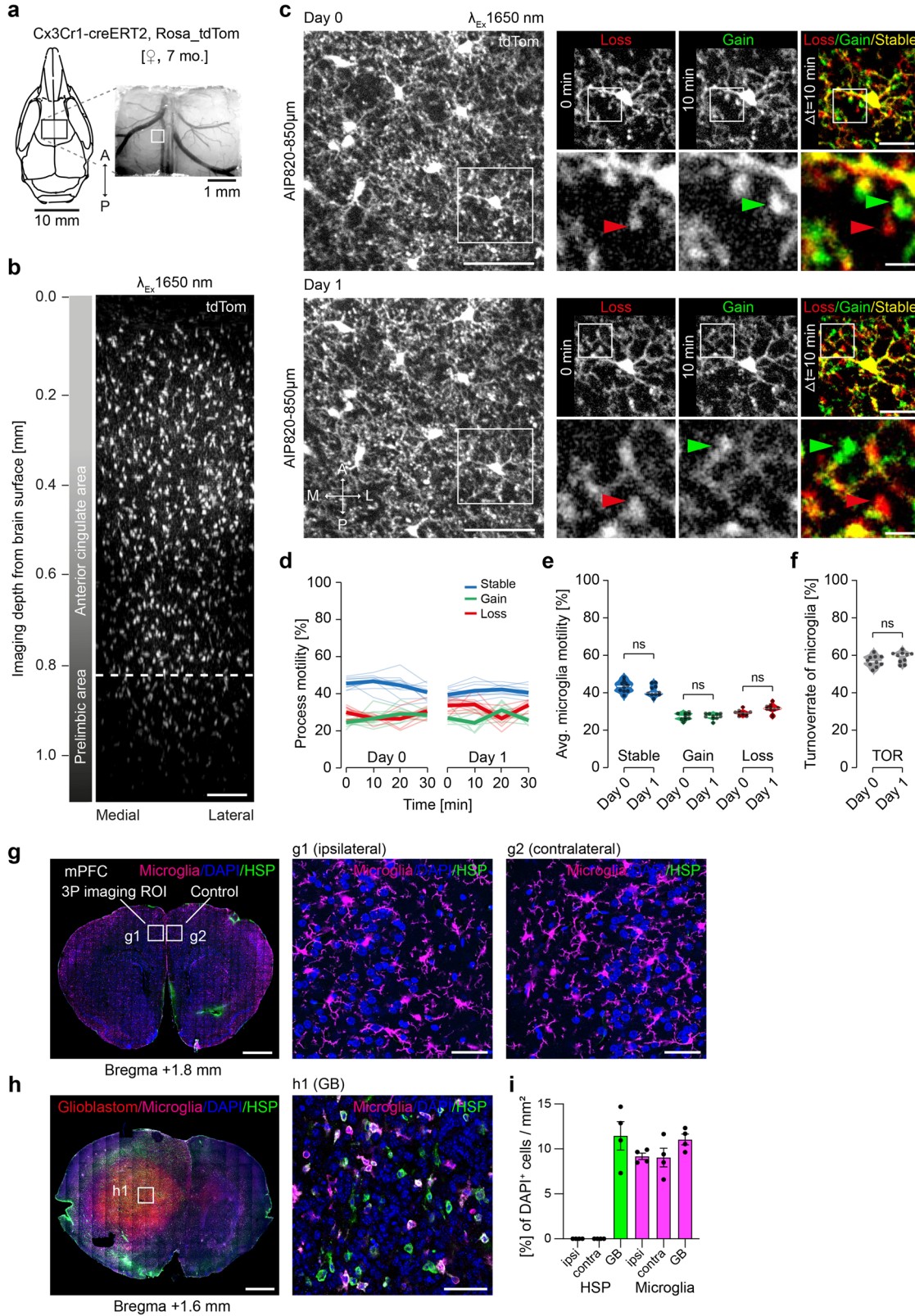

3P-imaging, we implanted cranial windows over the right somatosensory cortex in double transgenic Thy1-YFP-H::Cx3cr1-GFP mice (Supplementary Fig. 5a). These mice express YFP in a subset of excitatory neurons and GFP in microglia. With 2P-imaging at 920 nm microglia and neurons were visualized up to a depth of 600–700 µm, whereas 3P-imaging at 1300 nm excitation enabled the recording of microglia in the corpus callosum (800 µm deep) and neurons in dorsal CA1 of the hippocampus (Supplementary Fig. 5b, c, Supplementary Fig. 3c, d).

Red-shifted fluorescent indicators can be excited with less scattering at higher excitation wavelength light and promise improved imaging depth. We used *Cx3cr1-Cre^{ERT2}::Rosa26_tdTomato* transgenic mice[49] expressing tdTomato upon Tamoxifen-induction in microglia to carry out longitudinal

**Fig. 3 | Unchanged microglial fine process motility upon repetitive 3P-imaging in the mPFC. a** mPFC window positioning and image of the brain surface and vasculature (indicated ROI of z-stack shown in **b**) of a 7-month-old Cx3Cr1-creERT2-Rosa26tdTomato mouse. **b** 3D reconstruction of 212 $x$–$y$ frames from brain surface to 1060 μm below taken at a depth increment of 5 μm with 1650 nm excitation. Dashed line indicating in vivo imaging depth of microglial fine process motility recordings shown in **c**). Scale bar: 100 μm. **c** Average intensity projections of 9 frames (z-spacing: 3 μm) 850 μm deep of the very same microglia, recorded on two consecutive days. Right panels: Zoom of one microglia at 0, 10 min and in a color-coded overlay with red (lost), green (gained), and yellow (stable) microglia processes. Lower panels (Zoom from upper panel): A lost microglial fine process is marked with a red arrow, a gained with a green arrow at 0 min and 10 min, respectively. Scale bars: 50 μm (left panel), 20 μm (upper panel), 5 μm (lower panel). **d** Microglial process motility depicted as fraction of gained, lost and stable microglial processes over a period of 30 min ($n$ = 10 microglia, bold lines show mean, thin lines individual microglia). **e** Fractions of microglial fine processes' motility on two consecutive days.

Shown are individual data points and violin plots. One-way ANOVA with Šídák's multiple comparisons test. $F(2.331, 20.98) = 80.46$; adjusted $p$-values: $d0_{stable}$ vs. $d1_{stable} = 0.19$; $d0_{gain}$ vs. $d1_{gain} = 0.99$; $d0_{loss}$ vs. $d1_{loss} = 0.1$, from $n$ = 10 individual microglia, bold white lines indicate median. **f** Turnoverrate of microglial fine processes. Shown are individual data points and violin plots. Paired $t$ test, two-tailed, $p$ = 0.069, $n$ = 10 individual microglia, bold white lines indicate median. **g** Coronal section 1.8 mm anterior bregma of the previously 3P in vivo imaged ROI with tdTomato expression in microglia and immunohistochemical staining for heat shock protein (HSP) and DAPI. Scale bars: 1 mm (left panel), 50 μm (right panels). **h** Coronal section 1.6 mm anterior bregma through the tumor from a glioblastoma mouse model with immunohistochemical staining for HSP, Microglia (Iba1) and DAPI. Scale bars: 1 mm (left panel), 50 μm (right panel). **i** HSP positive cells and microglia as percentage of DAPI$^+$ cells per area in the 3P imaged ROI (ipsi.), contralateral hemisphere (cont.) and Glioblastom (GB). Mean of 4 ROIs each ±SEM (ipsi. = 1381 DAPI$^+$ cells, cont. = 1199 DAPI$^+$ cells, GB = 1248 DAPI$^+$ cells).

3P-imaging of microglia in the mPFC with 1650 nm excitation (Fig. 3a–c; Supplementary Videos 7, 8). We measured microglial fine process motility[50] for a period of 30 min of the very same microglia on two consecutive days (Fig. 3c–f; Supplementary Video 9). The morphology and position of microglia were unchanged comparing the two time-points (Fig. 3c). We measured 44.6 ± 4% stable fraction, 27.1 ± 4% gained and 28.4 ± 3% lost processes on the first imaging time-point. These numbers were comparable to the second time-point 24 h later (stable: 41.1 ± 2%, gain: 27.4 ± 4%, loss: 31.5 ± 5%) (Fig. 3d, e). Likewise, turnover rate of microglial processes was comparable between the two days (d0: 55.4 ± 4%, d1: 58.9 ± 2%) (Fig. 3f). To assess potential phototoxic or heating effects, we stained for heatshock protein (Hsp40) (Fig. 3g–i). While we detected 11.5% Hsp40-positive cells in a mouse model of glioblastoma, we did not find any Hsp40-positive cells in the ipsi- or contra-lateral side of the previously in vivo imaged mPFC region (Fig. 3i). Applying 3P-imaging in the mPFC, we repetitively measure the kinetics of microglial motility, underscoring the minimal invasiveness of our approach.

## 3P-imaging of astrocytic Ca²⁺-activity in mPFC in vivo

Astrocytes are highly active during different brain states, with patterns ranging from spontaneous Ca²⁺-events in discrete microdomains to larger somatic Ca²⁺-changes and coherent astroglial network activity changes[51,52]. Cortical astroglial activity has so far been mostly recorded in superficial cortical layers, although several studies have indicated that the molecular and cellular repertoire of astrocytes strongly differs between superficial and deep cortical layers as well as different brain regions[53,54]. Therefore, we aimed to image astroglial Ca²⁺-activity in deep mPFC layers. We used the mouse line *GLAST-CreERT2::GCaMP5g::tdTomato* that conditionally (Tamoxifen-induced) expresses the Ca²⁺-indicator GCaMP5g and the reporter protein tdTomato in astrocytes in a Cre-dependent manner after tamoxifen induction[55,56]. We imaged mPFC through a cortical cranial window in head-fixed, anesthetized, 4-month old mice (Fig. 4a). Robust and specific tdTomato reporter expression throughout all cortical layers was confirmed by post-hoc immunohistochemistry (Fig. 4b). We imaged astrocytes at 1300 nm excitation up to 1.2 mm deep from the brain surface, including the anterior cingulate and prelimbic areas (Fig. 4c, Supplementary Video 10). Ca²⁺-event analysis from deep cortical layers (1.0–1.2 mm from surface) showed that astrocytes in these layers displayed abundant spontaneous Ca²⁺ microdomains (Fig. 4d, e; Supplementary Video 11). The amplitudes, temporal kinetics and activity frequencies of these microdomains were similar to those reported for upper cortical layers in anesthetized mice using 2P-imaging[57]. In addition, we also recorded Ca²⁺-changes from astrocytic somata (Fig. 4f). In general, these transients were larger, longer and less frequent than microdomains, consistent with 2P-imaging data from upper layer astrocytes in anesthetized mice[58–60]. Hence, 3P-imaging enables recording of green- and red-fluorescent astrocytes at subcellular resolution in deep cortical layers. Moreover, our data suggest that, despite considerable layer-specific genetic heterogeneity, baseline Ca²⁺-activity in different cellular compartments is uniform across cortical layers in astrocytes.

## Imaging Ca²⁺-activity of neurons in the mPFC and dentate gyrus of awake head-fixed mice

Three-photon Ca²⁺-imaging of neurons has been performed in the cortex and hippocampus of mice, as well as in *Drosophila*[26,30]. In *Drosophila* we combined 3P-imaging with non-invasive mounting methods to carry out Ca²⁺-imaging in MB KCs in adult flies through the intact cuticle at cellular resolution confirming previous results (Supplementary Fig. 6, Supplementary Video 12, Supplementary Information 1). Ca²⁺-imaging in the mPFC of mice has been previously carried out through microprisms and GRIN-lenses[39–41]. These approaches differ by the size of accessible brain region and invasiveness (Supplementary Fig. 7). Here we employed three-photon Ca²⁺-imaging in the mPFC of awake head-fixed mice that expressed the Ca²⁺-indicator GCaMP6s in excitatory neurons (Fig. 5a, b). We used 1300 nm excitation to record somatic Ca²⁺-transients of VGluT2$^+$ neurons in layer V of the prelimbic area of the mPFC as deep as 1.4 mm below the brain surface (Fig. 5c; Supplementary Video 13). Regions of interest (ROIs) were imaged with framerates ≥10 Hz. We resolved and detected Ca²⁺-transients of more than 240 individual neurons in prelimbic layer V of the mPFC (Fig. 5d, e; Supplementary Video 14). The average Ca²⁺-transient amplitude was 4.95 ± 4.14 [% ΔF/F] with a mean half width of 3.24 ± 2.25 s (Fig. 5f, g). The spiking of each neuron was inferred from the corresponding Ca²⁺-traces (Fig. 5e) and a mean inferred spike frequency of 0.33 ± 0.18 Hz was calculated for VGluT2$^+$ neurons in prelimbic layer V of the mPFC in awake resting mice (Fig. 5h).

Ca²⁺-imaging in the hippocampus of awake mice is a powerful tool to correlate neuronal activity with behavior. The dentate gyrus (DG) can be reached with 2P-imaging through an implanted hippocampal window (Supplementary Fig. 8a–c). Since SBR of 3P-imaging was superior beyond 500 μm depth (Fig. 1g), we performed 3P-imaging of DG neurons in transgenic GCaMP6f mice through an implanted hippocampal window (Supplementary Fig. 8d, e; Supplementary Video 15). We imaged ROIs in DG with framerates ≥10 Hz and precisely resolved sparse Ca²⁺-transients in head-fixed mice running on a linear treadmill (Supplementary Fig. 8e). These experiments underscore the applicability of 3P-imaging for awake Ca²⁺-imaging in deep brain areas like the DG and the mPFC, and provide the basic Ca²⁺-transient parameters of layer V excitatory neurons in the mPFC.

## Discussion

Here we applied three-photon in vivo imaging to record from different cell types and with different modalities in the prelimbic area of the mPFC. We demonstrated high-resolution sub-cellular imaging at depths below 1000 μm from the brain surface. Specifically, we recorded neurons at cellular resolution up to 1600 μm depth. We repetitively imaged dendritic spines of prelimbic layer 5 neurons over a week. We measured the processes' motility of tdTomato-expressing microglia excited at 1650 nm in the prelimbic area of the mPFC. We carried out Ca²⁺-imaging in awake head-fixed mice to record Ca²⁺-transients from astrocytes and neurons in the prelimbic and infralimbic areas of the mPFC. Our data show that 3P in vivo imaging of prelimbic areas of the mPFC, without implantation of invasive GRIN lenses

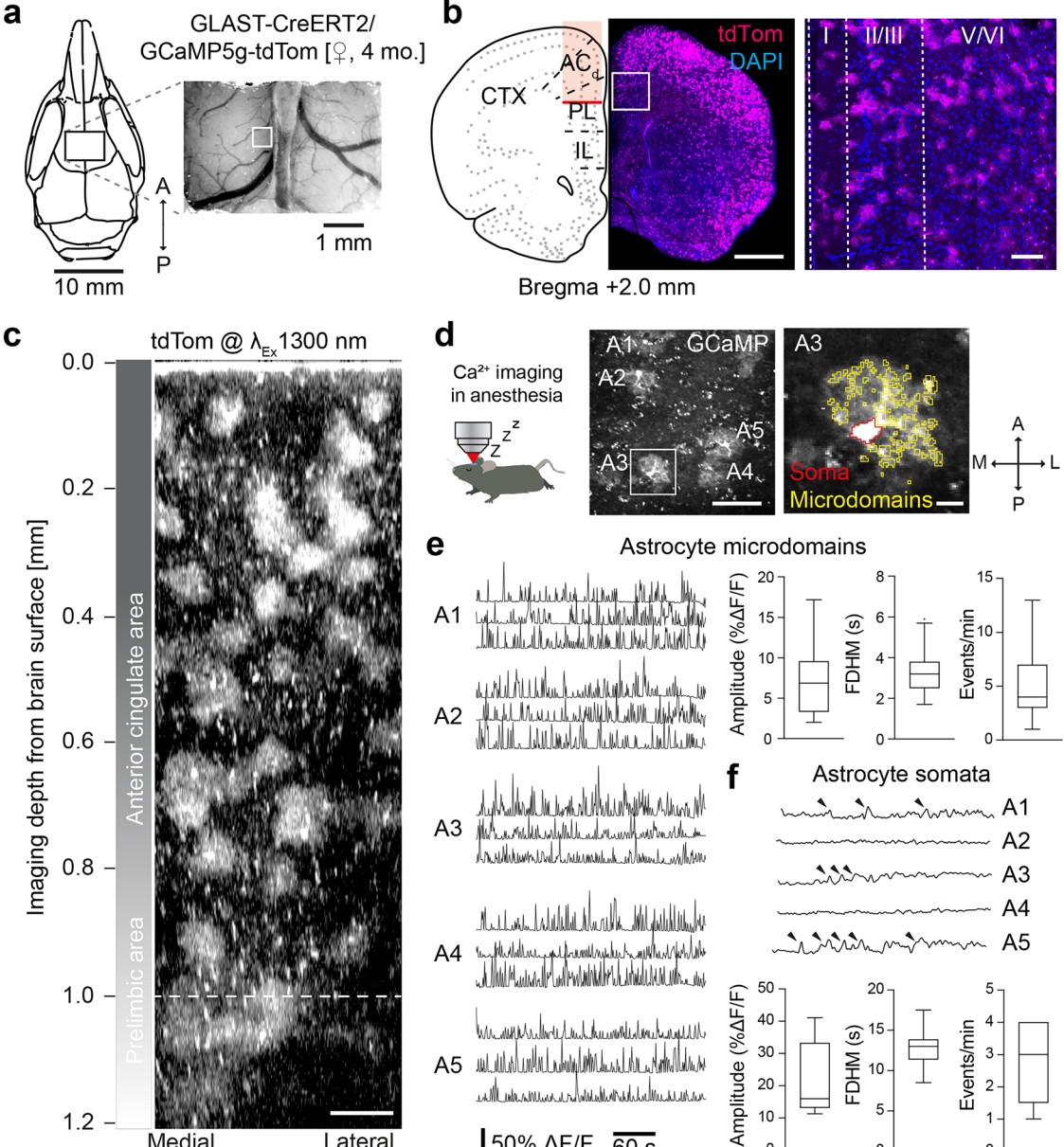

**Fig. 4 | Astrocyte Ca²⁺-imaging in deep cortical layers of the PFC. a** mPFC imaging window position on the skull and representative image of the brain surface and vasculature (left, indicated ROI of z-stack shown in **c**); scheme of the anatomical localization of imaging window in relation to the mPFC subareas (right). **b** Coronal section 2 mm anterior to bregma, and expression pattern of tdTomato in astrocytes in mPFC. Magnification and cellular reporter expression in different layers of mPFC. Scale bars: 1 mm, 100 μm. **c** 3D reconstruction of 406 x–y frames from brain surface to 1200 μm below, acquired at a depth increment of 3 μm. Scale bar: 100 μm. **d** In vivo 3P-imaging of GCaMP5g-positive astrocytes at 1000 μm below surface labeled A1-A5; left, zoom on A3 with automated ROI detection of the soma and active calcium domains. Scale bars: 100 μm, 10 μm. **e** Representative examples of calcium transients in astroglial microdomains recorded from A1-A5 with amplitude ($n = 265$ transients), full duration at half-maximum (FDHM, $n = 115$ transients) and event frequency ($n = 51$ microdomains). **f** Calcium changes in astroglial somata (recorded from cells A1-A5; transients are labeled by arrowheads) with amplitude ($n = 13$ transients), FDHM ($n = 8$ transients) and event frequency ($n = 5$ somata). Recordings in (**e**), (**f**) derive from $n = 10$ astrocytes in 2 mice, boxplot diagrams shown with median.

or microprisms, will be instrumental to elucidate the role of the mPFC in health and disease.

In the past, 3P in vivo imaging in rodents focused on reaching the hippocampus through the intact cortex[25]. Young mice (<2 months) only have a thin and weakly myelinated corpus callosum[61]. Therefore, the hippocampus is easily accessible even for less bright indicators used for Ca²⁺-imaging[26,62]. However, with age axons in the corpus callosum become highly myelinated, which increases light scattering, decreases excitation efficiency, and prevents image acquisition. Therefore, in older mice (>2 months) access to the hippocampus can instead be reached only from more lateral positions, where the excitation light does not need to pass through the thick

myelinated corpus callosum (Supplementary Figs. 2, 3, 5). Therefore, the dorsal hippocampal CA1 area is difficult to access with 3P-imaging in aged mice. This is especially the case for neurons expressing less brightly fluorescent GCaMP. The prelimbic area of the medial prefrontal cortex is positioned at a similar depth as the hippocampus, approximately 1 mm below the cortical surface. However, there is no myelinated fiber bundle separating the prelimbic area of the mPFC from the above-lying cortical areas. Indeed, we were able to visualize neurons 1700 μm deep in the mPFC, through a cranial window, without implanting GRIN lenses or microprisms previously used to access mPFC[43,44,63]. Furthermore, we recorded dendritic spines with a size of 1–2 μm at >1000 μm depth. The same dendrites and

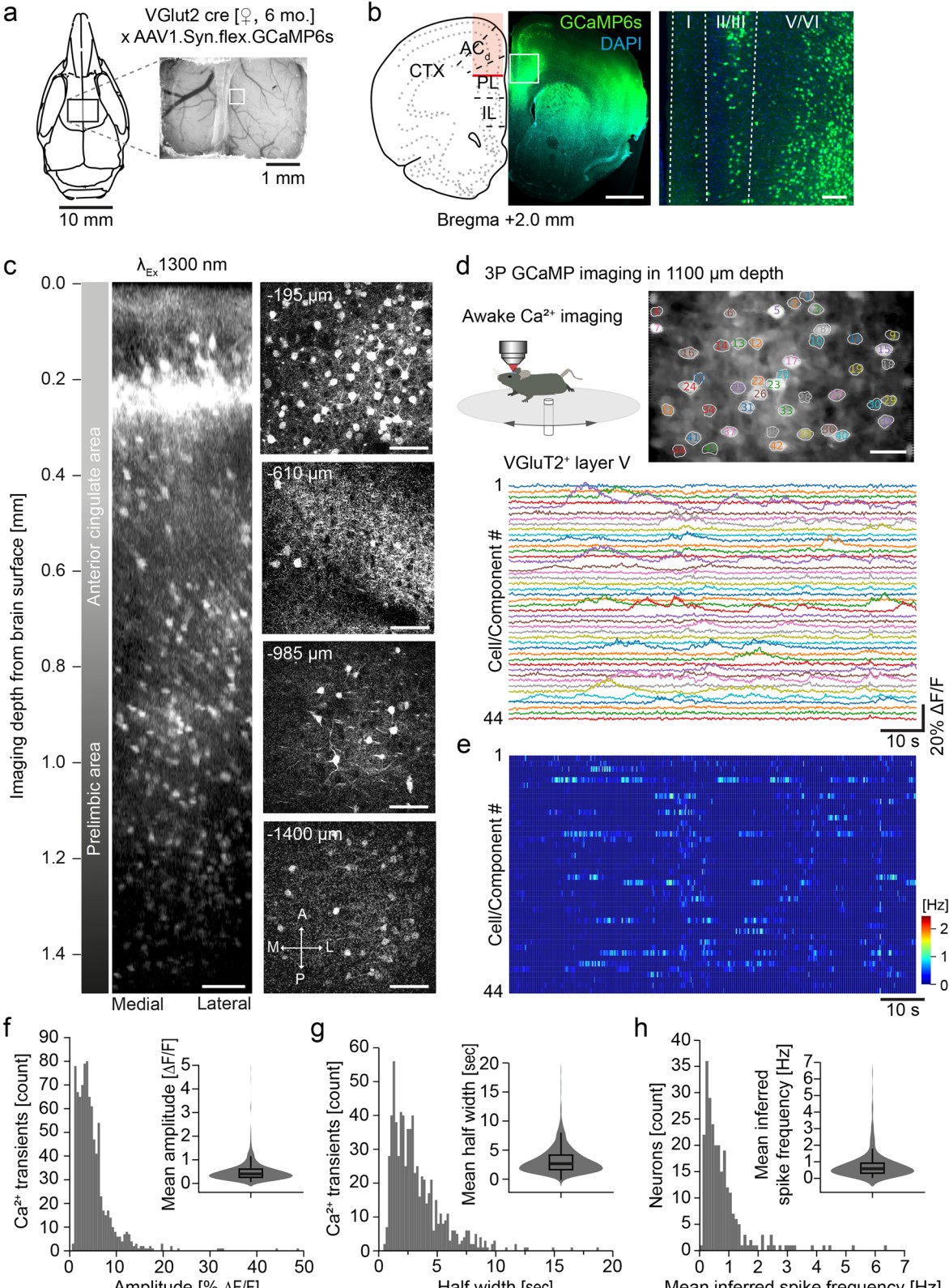

**Fig. 5 | Awake Ca²⁺-imaging of excitatory neurons in the mPFC. a** Schematic and exemplary picture of the cranial window placement in a vGlut2-Cre mouse to express GCaMP in glutamatergic neurons in the mPFC. **b** Schematic of a coronal section showing the imaged region in red (left). GCaMP6s expressing (green) neurons in the mPFC co-labeled with DAPI (blue). Overview (middle) and zoom (right) illustrating GCaMP6s expression in different cortical layers. Scale bars: 1 mm, 100 μm. **c** 3D reconstruction of a z-stack recorded from a depth up to 1420 μm below the brain surface (left). Exemplary images at various depths (right). Scale bars: 100 μm. **d** Schematic of 3P-imaging in an awake head-fixed mouse running on a circular disk (upper left). Average intensity projection with 43 ROIs of neurons in the mPFC at a depth of 1100 μm (upper right). Ca²⁺-recordings of the 43 labeled neurons (lower panel). Scale bar: 20 μm. **e** Inferred Ca²⁺-event rate (pseudo color coded) of the 43 neurons depicted in (**d**). **f–h** Number of Ca²⁺-transients over amplitude and mean Ca²⁺-transient amplitude (**f**). Number of Ca²⁺-transients over half width and mean half width (**g**). Number of neurons over mean inferred spike frequency and mean inferred spike frequency (**h**). (3 mice, 247 neurons, 882 Ca²⁺-transients, boxplot diagrams with median).

spines of layer 5 pyramidal neurons in the prelimbic area of the mPFC were repetitively imaged over 7 days. We were able to measure the spine density on basal dendrites of layer V neurons (0.41 ± 0.07 μm$^{-1}$), which represents a significant improvement compared with a prism-implantation approach that limits spine imaging to apical dendrites in layer I and are more invasive. In comparison to GRIN lens approaches that are fixed to one region of interest (stably implanted), our approach with a 4 × 3 mm window offers a larger brain region to be investigated. We would like to stress that no adaptive optics for wavefront shaping were necessary to resolve these tiny structures more than a millimeter deep. Adaptive optics have been used to improve the point spread function for sub-cellular resolution imaging in the hippocampus[28,31,64] and might also be beneficial for spine resolution improvement in the mPFC. However, adaptive optics are another costly device in the beam-path that potentially broaden the pulse width, which needs proper correction and adjustment. Our approach shows that sub-cellular resolution of dendritic spines, more than a millimeter deep, is achievable without adaptive optics in the mPFC at any age and in the hippocampus of young mice.

Microglia are innate immune cells in the brain and rapidly react to small disturbances in the brain parenchyma[10,13]. Staining of microglia with a synthetic dye (Dylight649) enabled 3P-imaging at >1000 μm in the juvenile mouse brain[65]. Furthermore, GFP-expressing microglia could be visualized through the skull[66]. Here, we used and open-skull approach and show 3P-imaging at 1650 nm excitation of tdTomato expressing microglia in adult mice up to a depth of 1200 μm in the prelimbic mPFC area, which might be important for future experiments in mouse models of aging and neurodegeneration. In addition, we provide kinetics of microglial fine processes at subcellular resolution >1000 μm deep below the brain surface for the first time. Consecutive imaging of the very same microglia on subsequent days revealed unchanged kinetics and unperturbed morphology. Heatshock protein stainings remained negative, suggesting sufficiently low excitation energy. The measured turnover rate (∼55%) was comparable to previous measurements at smaller depths in cortex and hippocampus[50,67–69], indicating that our imaging approach is well suited for long-term recordings of these sensitive cells.

Similar to microglia, astrocytes are glial cells of the brain that are highly sensitive to disturbances such as phototoxicity[70]. Astrocytic Ca$^{2+}$-imaging in anesthetized mice has been previously achieved in the corpus callosum applying 3P-imaging[28]. Here we perform astrocytic Ca$^{2+}$-imaging in the medial prefrontal cortex at >1000 μm depth. We are able to resolve somatic signals as well as astrocytic calcium microdomains in the prelimbic area of the mPFC. Interestingly, although several studies have shown that gene expression profiles of astrocytes differs strongly between superficial and deep cortical layers[53,54], our data indicate that astroglial baseline Ca$^{2+}$-activity—specifically amplitudes, temporal kinetics and activity frequencies—is uniform across all cortical layers under anesthetized conditions, suggesting that functional characteristics of astrocytes may differ from their genetic profile.

Awake Ca$^{2+}$-imaging in mice, correlating neuronal network activity with behavior, has become a major tool to investigate brain function. In the hippocampus Ca$^{2+}$-imaging and structural imaging have been established using implantation of tubes or GRIN lenses[11,12,71]. With 2P-excitation, recordings from the dentate gyrus and CA3 region of the hippocampus are possible[72–74]. Three-photon Ca$^{2+}$-imaging in the dentate gyrus improved the image quality compared to 2P-imaging (Supplementary Fig. 8). However, it should be noted that due to the slower pulse repetition rates of 3P lasers (2–4 MHz) in comparison with 2P lasers (∼80 MHz), a smaller number of neurons can be imaged at 5–10 Hz frame rates in the case of three-photon excitation (Fig. 5). Ca$^{2+}$-imaging in the mPFC of awake mice has been enabled in the past by implantation of microprisms[43,44,63] or GRIN lenses in combination with two- and one-photon excitation[63]. However, implantation of optics is usually at the expense of brain integrity. The implantation of a microprism limits access to superficial layers (1–3) of the medial prefrontal cortex. In contrast our approach is less invasive and gives access to deeper mPFC layers (3–5) that were previously out of reach. Therefore, 3P Ca$^{2+}$-

imaging in deep prelimbic and infralimbic areas of the mPFC will greatly advance our knowledge about neuronal ensembles and their relation to cognition in the future.

Here, we demonstrate sub-cellular three-photon in vivo imaging of dendritic spines, neurons, microglia and astrocytes in the prelimbic area of the mPFC of awake and anesthetized mice. We show that depth penetration to measure structural and functional dynamics of neurons and glial cells can be significantly improved by the application of 3P-imaging. As 2P-imaging in the past, 3P-imaging holds a great potential to revolutionize our understanding of the brain and especially the mPFC in the future.

## Materials and methods

### Resources table

| Reagent or Resource | Source | Identifier |
|---|---|---|
| Chemicals, Peptides, and Recombinant Proteins | | |
| 4-Methylcyclohexanol | Sigma-Aldrich | Cat. no.: 153095; CAS: 589-91-3 |
| 3-Octanol | Sigma-Aldrich | Cat. no.: 821859; CAS: 589-98-0 |
| Buprenorphine | Reckitt Benckiser | |
| Cefotaxime | Fisher Scientific | Cat. no.: 15535136 CAS: 63527-52-6 |
| Dexamethasone | Sigma-Aldrich | Cat. no.: D1756 CAS: 50-02-2 |
| Isoflurane | Virbac | Vetflurane® |
| Metamizol-Natrium 1 H$_2$O | Serumwerk Bernburg | Metapyrin CAS: 68-89-36 |
| Ketamin | Medistar | Ketamin 10% Inj.-Lsg. |
| Miglyol 812 Hüls Neutralöl | Caesar & Loretz | CAS: 73398-61-5 |
| Mineral oil | Sigma-Aldrich | Cat. no.: 330779; CAS: 8042-47-5 |
| Tamoxifen | Sigma-Aldrich | Cat. no.: T5648 CAS: 10540-29-1 |
| Xylacin | Bayer | Rompun® |
| Virus | | |
| pAAV.Syn.Flex.GCaMP6s.WPRE.SV40 | Addgene | CAS: 100845-AAV1 |
| Experimental Models: Organisms/Strains | | |
| Mus musculus: B6.Cg-Tg(Thy1-YFP)HJrs/J | The Jackson Laboratory | Strain #:003782 thy1-YFP-H |
| Mus musculus: B6.129P2(Cg)-Cx3cr1$^{tm1Litt}$/J | The Jackson Laboratory | Strain #:005582 CX$_3$CR-1$^{GFP}$ |
| Mus musculus: B6.129P2(C)-Cx3cr1$^{tm2.1(cre/ERT2)Jung}$/J | The Jackson Laboratory | Strain #:020940 Cx3cr1$^{CreER}$ |
| Mus musculus: Tg(Thy1-EGFP)MJrs/J | The Jackson Laboratory | Strain #:007788 Thy1-GFP line M |
| Mus musculus: B6.Cg-Gt(ROSA)26Sor$^{tm14(CAG-tdTomato)Hze}$/J | The Jackson Laboratory | Strain #:007914 Ai14 |

| | | |
|---|---|---|
| Mus musculus *Slc17a6*tm2(cre)Lowl/J | The Jackson Laboratory | Strain #:016963 Vglut2-ires-cre |
| Mus musculus C57BL/6J-Tg(Thy1-GCaMP6f)GP5.5Dkim/J | The Jackson Laboratory | Strain #:024276 GP5.5 |
| Mus musculus: B6;129S6-*Polr2a*Tn(pb-CAG-GCaMP5g,-tdTomato)Tvrd/J | The Jackson Laboratory | Strain #:024477 PC-G5-tdT |
| Mus musculus: Tg(Slc1a3-cre/ERT)1Nat/J | The Jackson Laboratory | Strain #:012586 GLAST-CreER |
| Drosophila melanogaster: VT030559-Gal4 in attP2 | Vienna Drosophila Resource Center | VDRC ID: 206077; FlyBase ID: FBtp0104353 |
| Drosophila melanogaster: UAS-GCamP6f, UAS-nls-mCherry | Gift from L. Scheunemann (Delestro et al.[76]) | FlyBase ID: FBal0281812, FBtp0071546 |
| Software and Algorithms | | |
| Fiji | https://fiji.sc/ | |
| IgorPro | https://www.wavemetrics.com/ | |
| Python | https://www.python.org/ | |
| Taro Tools | https://sites.google.com/site/tarotoolsregister/ | |

## Microscope

We built a three-photon microscope setup as shown in (Fig. 1a). Three-photon excitation is achieved with a wavelength-tunable excitation source from Spectra-Physics (NOPA, Spectra-Physics) pumped by a femtosecond laser (Spirit, Spectra-Physics). Dispersion compensation has been realized with a prism-based compensator from APE (FemtoControl, APE). More details on the group delay dispersion of the setup have been published elsewhere[75]. The laser repetition rate is maintained at 2 kHz. Laser intensity control was implemented by a lambda/2 plate (BCM-PA-3P, Thorlabs). A 50% underfilling of the objective's back aperture was achieved with lens pairs (f:+100 and f:-50; AC254-100-C-ML and LD1464-C-ML, Thorlabs). The 2P excitation source is a Ti:sapphire laser (Chameleon UltraII, Coherent). The laser repetition rate is at 80 MHz. Images are taken with a multiphoton microscope built of a multiphoton microscope base (EMB100/M, Thorlabs), a motorized stage module (MCM3000, Thorlabs), and an epi-fluorescence LED illumination beam path (LED: MCWHLP1 and trinocular with 10x eyepiece: MCWHLP1, both Thorlabs). Primary scan path optics for Bergamo microscope series with the 900–1900 nm coating are used. Two GaAsP photomultiplier tubes (PMT2100, Thorlabs) in the 14° detector module (BDM3214-3P, Thorlabs) are used for non-descanned detection. The ThorDaQ with 3P Mezzanine Card (TDQ3-ESP, Thorlabs) is processing the data from the photomultiplier tubes and realizes the synchronization with the excitation laser pulses from the NOPA-Spirit laser with single-pulse precision. ThorImage (Version 4.3, Thorlabs) is used to control imaging parameters. A 25x water immersion microscope objective with the numerical aperture of 1.05 (XLPLN25XWMP2, Olympus) is used. Green and red signals are separated by a 488 nm dichroic mirror (Di02-R488, Semrock) and 562 nm dichroic mirror (FF562-Di03). Then the GFP and third harmonic generated (THG) signals are further filtered by a 525/50 nm band-pass filter (FF03-525/50, Semrock) and 447/60 nm (FF02-447/60, Semrock) band-pass filter, respectively. Lateral movements are done with the 2D stepper motor (PLS-XY, Thorlabs), z-stacks are done using the built-in z device or a piezo focus device (PFM450, Thorlabs)[76–78].

## Mice

Mice were group-housed and separated by gender with a day/night cycle of 12 h. Water and food were accessible ad libitum. Mice of both sex were used for the experiments. The age ranged from 10 weeks to 16 months. All experiments were performed according to animal care guidelines complying with ethical regulations of the DZNE and were approved by the Landesamt für Natur, Umwelt und Verbraucherschutz of North Rhine-Westphalia (Germany) (81-02.04.2019.A084; 81-02.04.2018.A239; 81-02.04.2020.A059; 84-02.04.2017.A098). We have complied with all relevant ethical regulations for animal use. B6.Cg-TgN(Thy1-YFP-H)2Jrs mice (#003782) and B6.129P-CX3CR1tm1Litt/J mice (#005582) were purchased from The Jackson Laboratories. Thy1-YFP::CX3CR1-GFP mice were derived from a heterozygous crossing of CX3CR1-GFP and Thy1-YFP-H mice. Heterozygous Cx3cr1-creER/Rosa26_tdTomato/GFP.M mice were derived from crossing of Tg(Thy1-EGFP)MJrs/J mice (#007788), B6.129P2(C)-*Cx3cr1*tm2.1(cre/ERT2)Jung/J mice (#020940) and B6.Cg-*Gt(ROSA)26Sor*tm14(CAG-tdTomato)Hze/J mice (#007914) purchased from The Jackson Laboratories. Vglut2-Cre *Slc17a6*tm2(cre)Lowl/J mice (#016963), C57BL/6J-Tg(Thy1-GCaMP6f)GP5.5Dkim/J mice (#024276), B6;129S6-*Polr2a*Tn(pb-CAG-GCaMP5g-tdTomato)Tvrd/J mice (#024477) and Tg(Slc1a3-cre/ERT)1Nat/J mice (#012586) were purchased from The Jackson Laboratories.

## Fly stocks

Flies were raised in a 12 h/12 h light-dark cycle on a standard cornmeal-based diet at 25 °C, 60% relative humidity. All functional imaging experiments were performed on adult male flies 3–5 days after eclosion. The genotype of flies for imaging KC activity was +/+;UAS-GCaMP6f, UAS-nls-mCherry/+;VT030559-Gal4/+. VT030559-Gal4 is a driver for most of the KCs in the mushroom body. UAS-GCaMP6f encodes GCaMP6f while UAS-nls-mCherry encodes for mCherry with a nuclear localization signal, as reported in Delestro et al.[76].

## Tamoxifen injections

To induce Cre recombinase expression in microglial cells, adult Cx3Cr1-CreER mice were injected intraperitoneally (i.p.) with 0.1 mg/g body weight tamoxifen for five consecutive days. Tamoxifen was dissolved in miglyol (Miglyol 812 Hüls Neutralöl) and administered in a volume of 5 μl/g body weight. To induce expression of GCaMP5g-tdTomato in GLAST-Cre::GCaMP5g-tdTomato-loxP mice, mice were injected i.p. at 5 μl/g with an emulsion of 20 mg/ml tamoxifen (Sigma) in sunflower oil:ethanol mix (Sigma) for five consecutive days.

## Cortical window preparation

Cortical window surgery was performed four weeks before imaging. According to each animal protocol, mice were either anesthetized with isoflurane (induction, 3%; maintenance, 1–1.5% vol/vol; Virbac), or with an i.p. injection of ketamine/xylazine (0.13/0.01 mg/g body weight). Body temperature was maintained with a heating pad at 37 °C. Mice received buprenorphine (0.1 mg/kg; subcutaneously (s.c.), Reckitt Benckiser), dexamethasone (0.2 mg/kg; s.c., Sigma) and cefotaxime (2 g/kg; s.c., Fisher Scientific) shortly before the surgery. After fixation in a stereotaxic frame, the skin was removed under sterile conditions and a craniotomy above the prefrontal cortex (3×4 mm) or above the right somatosensory cortex (4 mm diameter) was created with a dental drill. The dura was carefully removed. The brain surface was rinsed with sterile saline, and adequate #1 coverslips (4 mm diameter or 3×4 mm) were sealed into the craniotomy with dental cement. For head-fixation during in vivo imaging a headpost (Luigs & Neumann) was cemented adjacent to imaging window. For analgesia, buprenorphine (0.1 mg/kg s.c.) was injected three times daily and Meta-mizol was (200 mg/kg) was applied to the drinking water for 3 consecutive days.

## Hippocampal window preparation

To image dentate gyrus granule neurons, a hippocampal window was implanted above the right dorsal hippocampus. Anesthesia was established with an i.p. injection of ketamine/xylazine (0.13/0.01 mg/g body weight).

Additionally, buprenorphine (0.05 mg/kg s.c.) was injected shortly before the surgery. After surgical tolerance was reached, mice were placed into a stereotactic frame, and part of the skin above the skull to access the cortex was removed with surgical scissors. The eyes were protected with eye-ointment to prevent drying of the eyes. The skull bone was rinsed with PBS and the periost was carefully removed. A three millimeter circular hole was drilled with a dental drill into the skull bone. Subsequently, the cortical brain tissue above the dorsal hippocampus was carefully aspirated and a circular metal tube (3 mm diameter, 1.8 mm high) sealed with a coverglass (3 mm diameter) that was glued with UV-curable adhesive to the bottom of the metal tube, was inserted. The tube was glued to the skull with UV-curable dental cement and a metal bar was placed next to the hippocampal window to enable repetitive positioning of the mouse under the microscope[12].

For head-fixation during in-vivo imaging a headpost (Luigs & Neumann) was cemented adjacent to the hippocampal window. Analgesia was carried out with buprenorphine injections (0.05 mg/kg s.c.), three times daily for 3 consecutive days. In-vivo imaging started after 4 weeks of recovery.

### Spinal cord window implantation
A female 12-week-old GFP-M mouse was administered buprenorphine (Temgesic, s.c, 0.1 mg/kg) and anaesthetized with ketamine-xylazine (i.p, 100 mg and 10 mg/kg respectively). Fur was shaved and skin sterilized with sequential iodine and ethanol swabs before transferring the mouse to a self-regulated heating pad at 37 °C. An incision was made over thoracic spinal cord. Muscle was retracted and a dorsal laminectomy of thoracic spinal level 12 was performed. Dura was removed using forceps (Dumont 5) and angled spring scissors (FST 15010-09). The vertebral column was stabilized using a multi-joint frame and adson forceps. A 3×4 mm coverslip was secured using UV-cured adhesive (Norland Products) and dental cement.

### Viral injections
Mice were anesthetized with an i.p. injection of ketamine/xylazine (0.13/0.01 mg/g body weight). After fixation in a stereotaxic frame and skin incision (5 mm), placement of the injection was determined in relation to bregma and a 0.5 mm hole was drilled through the skull. Stereotactic coordinates were taken from Allen brain reference atlas version 1 (2008). For $Ca^{2+}$ imaging of mPFC layer 5 excitatory neurons 1 µl pAAV.Syn.-Flex.GCaMP6s.WPRE.SV40 (Addgene) was injected bilaterally into mPFC of VGlut2-cre mice (+0.9 mm anterior-posterior, ±0.05 mm lateral and −1.4 mm ventral with a rostral/caudal 30° angle) at 0.1 µl/min, using a UltraMicroPump, 34 G cannula and Hamilton syringe (World Precision Instruments, Berlin, Germany). After surgery buprenorphine (0.05 mg/kg) was administered three times daily for 3 days. mPFC window surgery followed two weeks after AAV injection.

### In vivo imaging
Mice were anesthetized with isoflurane (1–1.5% vol/vol) and head-fixed under the microscope on a heating pad at 37 °C. Awake in vivo imaging was performed with habituated head-fixed mice on a rotating disc or on a linear treadmill. Mice were trained to run head-fixed on the linear treadmill. Velocity was read out by a rotation sensor, and the mouse's actual position was computed with the help of a reflection light barrier. In addition, mice received one automatic liquid reward per lap (2 m) at a fixed location.

### Image acquisition
Deep overview in vivo z-stacks were recorded with depth increments of 1–10 µm, 0.2–0.65 µm/pixel resolution and 2–3 µs pixel dwell time. Z-stacks (spanning 10–30 µm) of dendritic spines on basal dendrites of mPFC LV/VI neurons were imaged in 900–1100 µm depth with 1 µm depth increments, 0.15 µm/pixel resolution and 2 µs pixel dwell time. For the measurement of microglial fine process motility, z-stacks of individual microglia were imaged with 2–3 µm depth increments, 0.16 µm/pixel resolution, 2 µs pixel dwell time and with 5–10 min time-intervals for a period of 30 min. Timelapse imaging of astrocytic $Ca^{2+}$-activity in mPFC in vivo was performed at 3 Hz frame rates with 1 µm/pixel resolution and 2 µs pixel dwell time. Imaging the $Ca^{2+}$-activity of layer 5 excitatory neurons in mPFC of awake mice was performed at ≥10 Hz frame rates with 1–2 µm/pixel resolution and 2 µs pixel dwell time. Recordings of the $Ca^{2+}$-activity of dentate gyrus granule cells in the hippocampus of awake mice was performed at 5–10 Hz frame rates with 0.5–1.5 µm/pixel resolution and 2 µs pixel dwell time. Spinal cord in vivo z-stacks were recorded with depth increments of 3 µm, 0.25 µm/pixel resolution and 2–3 µs pixel dwell time.

### $Ca^{2+}$ imaging of the Drosophila MB KCs with intact cuticle
Male adult flies were briefly anesthetized on ice, and then placed on a chilled aluminum block. The posterior part of the head capsule was positioned upward, and attached to the edge of a glass coverslip using UV-curable glue (NOA 68, Norland Products Inc.). As the KC cell bodies are closer to the cuticle on the posterior side compared to the dorsal side of the head, this mounting method could effectively expose the cell bodies, and reduce light scattering through a thinner layer of fat body. Then, the glass coverslip with the head-fixed fly was positioned carefully in and fixed to the imaging chamber. The flies were then imaged with the 3P microscope setup described here and equipped with a 25× Olympus XLPlan N WMP2 (1.05 NA, 2.0 mm WD) water-immersion objective. Both GCaMP6f and nls-mCherry were excited at 1300 nm. Bandpass filters for 525/550 nm and 593/620 nm were used for detecting the GCaMP and mCherry fluorescence signals, respectively. Individual monomolecular odors were used to stimulate the olfactory responses of KCs. Odor stimuli were delivered to the fly using a 220 A olfactometer (Aurora Scientific). The odors 3-Octanol (OCT) and 4-Methylcyclohexanol (MCH) were diluted at 1:100 and 1:10 in mineral oil, respectively. Upon delivery to the fly, the olfactometer further diluted the odor in air, resulting in the final dilution of 1:1000 for OCT and 1:100 for MCH. Three trials of 3 s odor pulses of OCT, MCH, or mineral oil were delivered to the animal in a randomized order, with 35 s constant flow of pure air between odor exposures. After each imaging session, the fly can be detached from the coverslip and return to the standard food vial with other flies.

### Histology
Mice were deeply anesthetized with an i.p. injection of ketamine/xylazine and transcardially perfused with saline followed by 4% paraformaldehyde (PFA). Brains were taken out and coronal 70 µm slices were cut. Subsequently, after permeabilization (0.5% Triton-X100, 1 h), slices were incubated with a DNAJ4/HSP40 antibody (1:100, mouse serum, Santa Cruz Biotechnology, sc-100714) and an Iba1 antibody respectively in the glioma condition (1:1000, rabbit serum, Wako, 019-19741) in a blocking reagent (4% normal goat serum, 0.4% Triton 1%, and 4% BSA in PBS) over night at room temperature. After washing the samples three times with PBS, secondary antibodies were administered (Alexa Fluor 647, goat anti-mouse, A21235, 1:400 and Alexa Fluor 488, goat anti-rabbit, 1:400, A11008) in 5% normal goat serum/BSA and incubated for 2 h at room temperature. During the last 15 min of incubation DAPI was added (5 mg/ml, 1:10000). Afterwards, slices were washed three times with PBS, mounted with Dako Mounting Medium, and covered with a glass cover-slip.

### Confocal Imaging
For the visualization of the different expression pattern and anatomical orientation, coronal slices were imaged with an LSM800 microscope. DAPI- (EX: G 365, Dichroic: FT 395, EM: BP 445/50), GFP- (EX: BP 470/40, Dichroic: FT 495 EM: BP 525/50), YFP- (EX: BP 500/20, Dichroic: FT 515, EM: BP 535/30) and tdTomato- (EX 561/10, Dichroic: 573, EM: 600/50) filter-sets were used. Large overview images were generated by stitching average intensity projections of x/y/z tile-scans with depth increments of 5–10 µm and 0.2–1.0 µm/pixel.

## Data analysis

**Calcium imaging of neurons**. For the analysis of neural activity in the mPFC and DG, a combination of tools taken from the python toolbox for calcium data analysis CaImAn[79], the python toolbox for spike inference from calcium data Cascade[80] and Igor Pro as well as custom-written code in python were used. Motion correction of the imaging recording was performed using the CaImAn-implementation for rigid-body registration. Detection of cell bodies and source-separation was performed using the CaImAn algorithm based on constrained non-negative matrix factorization[81] and the raw calcium signal was extracted from the ROIs. For DG recording, the ROIs were selected manually. $\Delta F/F$ was calculated using following equation: $(F - F_0)/F_0$ where $F_0$ is the minimum 8th quantile of a rolling window of 200 frames. Depending on the sampling rate of the recording, for spike inference the cascade algorithm trained to either the datasets Global_EXC_6Hz_smoothing200ms, Global_EXC_7.5Hz_smoothing200ms or Global_EXC_10Hz_smoothing50ms was used. For peak detection and determination of amplitude and half-width of decay we used Taro tools (https://sites.google.com/site/tarotoolsregister/).

## Calcium imaging of astrocytes

Calcium imaging data were stabilized using a custom-written Lucas-Kanade algorithm[82] in Matlab R2018a (MathWorks). Timelapse recordings were down sampled to 1 Hz framerates by 3x frame averaging. Regions of interests (ROIs) representing astroglial microdomains were defined by fluorescence changes over time in GCaMP5g-positive astrocytes using a custom-written macro in ImageJ 1.50i (modified from[57]). Time-lapse data for each ROI were normalized, smoothed, and peak candidates were detected with a hard threshold. Detection and classification of fluorescence peaks over time was performed with a custom-written algorithm in Python. Mean fluorescence data were first normalized by a robust z-score calculated per ROI over the whole time-lapse series. Normalized data were then smoothed with a Gaussian filter, and all maxima above the threshold were selected as peak candidates. Peak candidates were defined by their ROI and the timepoint of peak maximum. Peak amplitude and full duration at half-maximum (FDHM) were determined for each peak candidate. Each time-lapse series was plotted together with the respective video file for visual inspection and verification.

## Ca²⁺-imaging of the Drosophila MB

Image processing was conducted using Fiji, and data analysis was performed with custom codes written in python. For the MB calcium imaging, the 2D time series was first motion-corrected using the 'Template matching and slice alignment' plugin in Fiji, which performed slice registration with a selected landmark. Afterwards, the cell bodies were identified using the nls-mCherry signal and the segmentation function of the 'Trackmate' plugin in Fiji, followed by manual adjustment. To quantify the neural responses along the time series, we extracted the fluorescence intensity of individual KCs for both the nls-mCherry and the GCaMP6f channels. The baseline fluorescence level ($F_0$) was calculated by averaging the 17 frames before the first odor delivery for each channel. $dF/F_0$ for both channels were then computed using $(F_t - F_0)/F_0$, respectively. The axial-brain-motion-induced fluorescence change at the GCaMP channel was corrected with the use of the $dF/F_0$ from the nls-mCherry channel. To be considered as responsive to a particular odor, the peak $dF/F_0$ within 10 frames after that odor release should be above the responding threshold (2STD). Only those KCs that respond to the same odors across all three trials are considered as responding units.

## Dendritic Spines

To remove noise in dendritic spine image data, a median filter with a kernel size of 3×3 pixels was applied. The image data were subsequently stabilized using an optical flow method based on an iterative Lucas-Kanade solver[83,84] from the open-source Python image processing library Scikit-image[85]. The measurement of dendritic spine density was conducted by experienced human analyzers, who counted individual spines in each image stack and normalized the spine counts to the corresponding dendrite lengths (see, e.g.[19,50]).

## Microglia

Microglial fine process motility was measured with a custom analysis pipeline "MotilA" written in Python[86]. First, noise was removed by applying a median-filter with a kernel size of 3×3 pixels. The individual consecutive time lapses of one mouse were maximum intensity projected along the z-axis and rigidly registered on each other using subpixel image registration by cross-correlation[82] provided by the open-source Python image processing library Scikit-image[85]. Subsequently, the registered z-projections were binarized by thresholding the pixels in each projection using Otsu's method[87]. The thresholding was preceded by contrast-limited adaptive histogram equalization (CLAHE)[88], also provided by the Scikit-image library, in order to enhance the contrast and, thus, the thresholding in each z-projection. Small blobs of less than 100 pixels were removed from the binarized images. We then calculated the temporal variation $\Delta B(t_i)$ of each binarized image by subtracting the binarized image $B(t_{i+1})$ at time point $t_{i+1}$ from two times the binarized image $B(t_i)$ at time point $t_i$: $\Delta B(t_i) = 2 \times B(t_{i+1}) - B(t_i)$ for $i = 0, 1, 2, …, N-1$, where $N$ is the total number of all time lapse time points. Pixels in $\Delta B(t_i)$, that have the value 1, were categorized as stable pixels, whereas pixels with the value $-1$ were categorized as gained pixels, and pixels with the value 2 as lost pixels. The microglial fine process motility was assessed by calculating the turnover rate (TOR) as the ratio of the number of all gained pixels $N_g(t_i)$ and all lost pixels $N_l(t_i)$ divided by the sum of all pixels: $TOR(t_i) = (N_g(t_i) + N_l(t_i)) / (N_s(t_i) + N_g(t_i) + N_l(t_i))$, where $N_s(t_i)$ is the number of all stable pixels. The average turnover rate $\overline{TOR}$ was calculated by averaging $TOR(t_i)$ over all $N-1$ time lapse time points.

## Signal to background ratio (SBR)

Signal to background ratio was calculated for each frame individually by manually placing and measuring brightest signal- and lowest background ROI.

## Cell counting

Single plane images of immunohistochemical stained brainslices were threshold adjusted, binarized and underwent watershed-algorithm based separation. Analysis was performed with Fiji (analysis of particles). >1000 DAPI⁺ cells were counted in four ROIs for each area.

## Statistical analysis

Quantifications of microglial motility and subsequent statistical analysis, and graph preparation were carried out using GraphPad Prism 9 (GraphPad Software Inc, La Jolla, CA, USA). To test for normal distribution of data, D'Agostino and Pearson omnibus normality test was used. Statistical significance for groups of two normally distributed data sets paired or unpaired two-tailed Student's $t$ tests were applied. One-way ANOVA with Šídák's multiple comparison test was performed on data sets larger than two, if normally distributed. If not indicated differently, data are represented as mean ± SEM. Figures were prepared with Illustrator CS5 Version 5.1 (Adobe) and IgorPro.6.37.

Boxplots were used to visualize the distributional characteristics of the data (Fig. 4e, f; 5f–h). Each boxplot displays five key summary statistics: the median (50th percentile), the lower quartile (Q1, 25th percentile), the upper quartile (Q3, 75th percentile), and the "whiskers" indicate minimum and maximum values. The central line within each box indicates the median, while the box spans the interquartile range (IQR = Q3-Q1), representing the middle 50% of the data. Violin plots visualize both the distribution and the density of the data (Fig. 3e, f; Fig. 5f–h). Each violin plot combines a boxplot with a kernel density estimation (KDE), providing information on the central tendency, spread, and shape of the distribution. Bold central line indicates the median. The width of the violin at each vertical position reflects the estimated probability density of the data at that value.

## Reporting summary

Further information on research design is available in the Nature Portfolio Reporting Summary linked to this article.

## Data availability

The authors declare that the data supporting the findings of this study are available within the paper, the methods section, and the Supplementary Data 1 file. Raw data of imaged z-stack videos can be downloaded from DRYAD (https://doi.org/10.5061/dryad.tqjq2bw90).

## Code availability

The newly developed code used for microglia motility analyses (MotilA[86]; https://doi.org/10.5281/zenodo.15175054) in the manuscript has been deposited openly accessible in a GitHub repository (https://github.com/FabrizioMusacchio/MotilA) along with a detailed documentation. Tutorial scripts to facilitate testing and understanding of the workflow are provided. Sample datasets are deposited on DRYAD (https://doi.org/10.5061/dryad.tqjq2bw90).

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

## Acknowledgements

The authors would like to thank Christof Moehl for providing support on astrocytic Ca$^{2+}$-imaging data analysis. This work was supported by the DZNE, and grants to M.F. by the European Union ERC-CoG (MicroSynCom 865618), ERA-NET Neuron (MicroSchiz) and the German research foundation DFG (SFB1089 C01, B06; SPP2395). This work was further supported by grants to G.C.P. from the German Science Foundation (Deutsche Forschungsgemeinschaft DFG, FOR 2795 "Synapes Under Stress", PE1193/6-2), ERA-NET (MICRO-BLEEDs and TACKLE-CSVD), and the Fondation Leducq (Transatlantic Network of Excellence 23CVD03). G.C.P. is a member of the DFG excellence cluster ImmunoSensation2. This work was also supported by a grant of the German Science foundation (FOR 2705, TA 265/5-2) to G.T., by the iBehave network to G.T. & M.F. and CANTAR (CANcerTARgeting) network to F.N. (funded by the Ministry of Culture and Science of the State of North Rhine-Westphalia; the funders had no role in study design, data collection, and interpretation, or the decision to submit the work for publication). F.N. received funding from the Mildred-Scheel School of Oncology Cologne-Bonn.

## Author contributions

F.F. and M.F. prepared figures and wrote the manuscript with input from all authors. F.F., F.C.N. carried out surgeries and recorded the data. F.F. and F.M. established the setup together with S.K. and H.F. M.a.M. recorded data and analyzed Ca$^{2+}$-imaging data. S.P., M.o.M., E.A.G., and M.i.M. carried out measurements. B.S. and E.B. carried out spinal cord surgeries and recordings, partly together with F.F. They wrote parts of the manuscript together with F.B. A.S.L., I.C.W.C. carried out *Drosophila* measurements, prepared a figure and wrote parts of the manuscript together with G.T. N.R. performed astrocytic measurements together with F.F. and performed astroglial calcium analyses. N.R. and G.C.P. prepared a figure and wrote parts of the manuscript. S.L. carried out autocorrelator experiments together with F.M. G.C.P., G.T., F.B., and M.F. planned the project and supervised the work.

## Funding

## Competing interests

The authors declare no competing interests.
