## [Transparent Peer Review file · Communications Biology]

Three-photon in vivo imaging of neurons and glia in the medial prefrontal cortex with sub-cellular resolution

Corresponding Author: Professor Martin Fuhrmann

This manuscript has been previously reviewed at another journal. This document only contains information relating to versions considered at Communications Biology.

Version 0:

Reviewer comments:

Reviewer #1

(Remarks to the Author)

In this paper, Fuhrmann and colleagues demonstrate that three-photon imaging enables the imaging of several aspects of neural cell structure and dynamics in the deep-laying areas of the murine prefrontal cortex (with some images from the hippocampus and *Drosophila* brain). They describe in some detail their setup (which is semi-commercial) and some adjustments to the excitation laser pulses, e.g., via pulse compression. Several parameters are compared to two-photon imaging as the reference method, which, however, cannot penetrate many deeper-seated parts of the medial prefrontal cortex. The paper's novelty claim is that this represents the first application of three-photon microscopy to deep prefrontal cortex areas, which is of substantial interest in behavioral paradigms and disease models.

Overall, I found the paper technically sound and easy to read. My problem is that it contains hardly any surprising results – while it is nice to see that three-photon microscopy can reach the deep prefrontal cortex, I think this could be deduced from first principles and prior literature. Indeed, three-photon imaging has long been demonstrated to enable deeper imaging penetration of brain tissue in substantial prior work by other labs (see, e.g., Horton Nat Phot 2013; most recently, Thornton et al., Nat Neuro 2024, but also miniature 3P systems, Klioutchnikov Nat Meth 2020).

Thus, while I think the paper contains an interesting application of a well-documented technique, I would argue that this does not justify a stand-alone report – after all, “deep imaging” is the established rationale why three-photon microscopes or kits to build them are now sold by several commercial providers; hence any likely reader of this article would typically concur with its conclusion beforehand. Therefore, I see this more as an application “white paper” rather than a primary research report.

Minor comments to figures and movies

Figure 3c: It would be beneficial to include a supplementary video of the 30-minute time-lapse showing the motility of individual microglia at 850 μm depth

Supp. Fig. 3: Imaging of the spinal cord dorsal horn gray matter at ~ 400 μm depth. This might warrant some discussion, as there is a preprint by the Schaffer lab reporting deeper imaging (>500 μm ; doi: 10.1101/2024.04.04.588110), albeit this might depend on the criteria of when an image is considered “useful”.

Supp Fig. 5: What is the imaging depth for the Kenyon cells shown in Supp Fig. 5e?

Supp Fig. 4c: The YFP/GFP look-up table makes it difficult to distinguish overlapping microglia and neurons. What are the red signals in the three-photon panel?

Supp Video 5: The images at 750 μm and 1000 μm depth shown in Figure 2C don't align with the corresponding images in the video.

Supp Video 9: The background is grainy with high noise. It would be helpful to visualize astrocyte structures by overlaying the cytoplasmic tdTom expression with GCaMP5g signals.

Reviewer #2

(Remarks to the Author)

In this work, Fuhrmann et al. presented a new demonstration of applying three-photon (3P) in vivo imaging to achieve sub-cellular resolution in the medial prefrontal cortex (mPFC) of mice at unprecedented depths (up to 1700 μm). The mPFC is an important brain region for studying higher cognitive functions and has been of significant interest in neuroscience research. This is the first reported in vivo 3P imaging demonstration to achieve such depth in this area. This work demonstrated the imaging of both neuronal and glial activity, including dendritic spines, astrocytic Ca^{2+} -activity, and microglial processes, over extended periods with minimal invasiveness. The research highlights the advantages of 3P imaging over traditional two-photon (2P) methods, particularly in terms of depth penetration and imaging quality in deep brain structures.

Overall, I am very supportive of this work being published. The presented data is of high quality, and the demonstrations are generally convincing, with broad applications to various potential neuroscience problems. However, I have some technical comments that the authors need to address before final publication.

Major comments:

1. My biggest concern is to achieve deeper imaging in the work, the authors chose a lateral imaging position to avoid imaging through highly myelinated corpus callosum layers which typically start to appear within 1.5 mm if imaging through somatocortex and deeper region as demonstrated in many other 3P mouse brain imaging work.

As mentioned in the paper "Since it is difficult to penetrate the highly myelinated corpus callosum in adult mice (4 months old), we chose a lateral imaging position to circumvent it (Supplementary Fig. 2a)." It would be better to discuss

a. How this location is searched and identified? This could be helpful protocol for other people interested in imaging deeper without the concern of white matter.

b. Since it's less common to see 3P mPFC imaging results, it will be valuable to also learn the typical 3P imaging depth through white matter?

c. What if a specific biological question requires not avoiding the highly myelinated corpus callosum, can the authors also show how deep 3P imaging can reach in their case, when imaging mPFC, while imaging depth of 1.7 mm without going through mPFC is impressive in value, imaging depth of the case going through mPFC is also valuable to report.

2. The authors reported the spine density at various depth. This is impressive since it was previously reported this value can fluctuate due to animal motion. Particularly in this case, the demonstrated imaging was performed with head-fixed, awake, treadmill running mouse. What's the stability of the spine imaging, does this impact the reported values? For example, there could be some spines be out of focus or becoming more visible over a few minutes to hours imaging window due to animal motion. Can the authors comment on this?

3. My other big concern is related to this claim "we did not use any adaptive optics to correct for wavefront distortions" and related ones "We would like to stress that no adaptive optics for wavefront shaping were necessary to resolve these tiny structures more than a millimeter deep." And "Our approach shows that subcellular resolution of dendritic spines, more than a millimeter deep, is achievable without adaptive optics".

I highly doubt that this was due to the fact that the authors don't have to deal with highly myelinated corpus callosum which is the major challenge of high-resolution imaging of dendritic spines at depth and the need for adaptive optics there. The authors need to fairly compare with similar cases in other adaptive optics work, where many of them have gone through the white matter layer.

I suggest editing this sentence and other related discussion with adaptive optics to make a fairer comparison and claims. If the authors still manage to visualize sharp and clear spine imaging through white matter (highly myelinated corpus callosum), then this data should be provided and support this claim.

4. "Interestingly, although several studies have shown that the genetic composition of astrocytes differs strongly between superficial and deep cortical layers 52,53, our data indicate that astroglial Ca^{2+} -activity, at least under anesthetized conditions, is uniform across all cortical layers."

a. This sentence was not clear. What's the specific difference the authors refer to for astrocytes with different genetic composition?

b. What does it mean when the authors claim uniform activity across all cortical layers, is it about the distribution of certain activity-related parameters or what? Please clarify.

5. Some comments on figure 1f and g in terms of SBR and power used:

a. Figure 1f, the initial power of 3P before 0.8 mm is lower than 2P, what's the rationale there? Can the authors show the 2P and 3P power curve (signal versus power to validate a pure 2P and 3P response)?

b. Why the maximum power of 2P and 3P controlled to be the same 200mW? They have different wavelengths in use, so heating impact should be different. Thus, the maximum power control doesn't have to be the same.

c. Why the initial SBR of 2P is higher than 3P? This needs explanation since 3P normally has higher SBR. Please refer:

- Figure 2b in Wang et al. "Quantitative analysis of 1300-nm three-photon calcium imaging in the mouse brain" eLife 2020

- Figure 3B in Takasaki et al. "Superficial bound of the depth limit of two-photon imaging in the mouse brain" eNeuro 2020

- Figure 2B in Akbari et al. "Imaging deeper than the transport mean free path with multiphoton microscopy" Biomed. Optics Express 2022

6. All statistical analysis in figures need to complement with clear explanation in the caption on the meaning of the mid-line, error bars: are they mean, median, std? This applies to figure 2d, 3d, 3e, 3f, 4f, 5f-h and some supplementary figures.

Reviewer #3

(Remarks to the Author)

In this manuscript, the authors demonstrated imaging of neurons and glial cells in the medial prefrontal cortex (mPFC) of mice using 1300-nm and 1650-nm three-photon microscopy. The authors established that three-photon imaging enables deep tissue imaging without invasive optical elements like GRIN lenses or microprisms, which are required by other optical imaging methods. The high variety and quality of experimental data make this work a valuable addition to the field. The authors adopted meticulous and thorough approaches towards experiments, and their data on optimizing optomechanical parameters for three-photon imaging provides valuable reference information for improving three-photon microscopy as a practical imaging modality. Overall, I strongly recommend its publication.

The authors demonstrated three-photon imaging of neurons and astrocytes up to 1.6 mm depth in the mouse mPFC, a region traditionally difficult to access optically. They rigorously compared three-photon to two-photon imaging depth, quantifying metrics such as signal-to-background ratio and required laser power. For chronic imaging, they impressively demonstrated the ability to resolve dendritic spines at depths beyond 1000 μm in mPFC. They also observed microglial process motility and studied calcium dynamics in astrocytes, finding uniform calcium activity patterns across cortical layers despite known genetic heterogeneity. The calcium imaging results in both mPFC neurons and dentate gyrus through a hippocampal window is also quite impressive. The technical details of the work seem quite solid to me, and I do not have any particular comment on them.

One area where the manuscript could be improved is in comparing the practical tradeoffs between three-photon imaging and alternative techniques to image mPFC using microprisms or GRIN lenses. While the authors mention that their 4 x 3mm window offers a larger brain region to investigate compared to stably implanted GRIN lenses, a more detailed comparison of field-of-view sizes, imaging depths, and achievable resolution would be valuable. The authors note that microprism approaches are limited to superficial layers (1-3) of the mPFC, while their technique accesses deeper layers (3-5), but there is no fundamental reason why the microprism could not be inserted deeper. Also, these microprisms convert imaging breadth to depth, and may lead to larger field-of-views. Therefore, the authors should have a more in-depth discussion on the comparison.

Last but not least, the work would be more impactful if the authors choose to publicize their raw imaging data through a publicly accessible data repository. I would encourage the authors to do so, even though it is not mandatory.

Reviewer #4

(Remarks to the Author)

Version 1:

Reviewer comments:

Reviewer #2

(Remarks to the Author)

All my previous comments have been well addressed by the authors. I appreciate their efforts in conducting additional

experiments with excellent results. I support its publication and believe it will be a valuable contribution to the three-photon brain imaging community.

Reviewer #3

(Remarks to the Author)

I would like to thank the authors for addressing my comments, and I would like to reiterate my support on the publication of the manuscript.

Reviewer #4

(Remarks to the Author)

The authors addressed my questions and comments.

Reply to Reviewers:

Reviewers' comments:

Reviewer #1 (Remarks to the Author):

In this paper, Fuhrmann and colleagues demonstrate that three-photon imaging enables the imaging of several aspects of neural cell structure and dynamics in the deep-laying areas of the murine prefrontal cortex (with some images from the hippocampus and *Drosophila* brain). They describe in some detail their setup (which is semi-commercial) and some adjustments to the excitation laser pulses, e.g., via pulse compression. Several parameters are compared to two-photon imaging as the reference method, which, however, cannot penetrate many deeper-seated parts of the medial prefrontal cortex. The paper's novelty claim is that this represents the first application of three-photon microscopy to deep prefrontal cortex areas, which is of substantial interest in behavioral paradigms and disease models.

Overall, I found the paper technically sound and easy to read. My problem is that it contains hardly any surprising results – while it is nice to see that three-photon microscopy can reach the deep prefrontal cortex, I think this could be deduced from first principles and prior literature. Indeed, three-photon imaging has long been demonstrated to enable deeper imaging penetration of brain tissue in substantial prior work by other labs (see, e.g., Horton *Nat Phot* 2013; most recently, Thornton et al., *Nat Neuro* 2024, but also miniature 3P systems, Klioutchnikov *Nat Meth* 2020).

Thus, while I think the paper contains an interesting application of a well-documented technique, I would argue that this does not justify a stand-alone report – after all, “deep imaging” is the established rationale why three-photon microscopes or kits to built them are now sold by several commercial providers; hence any likely reader of this article would typically concur with its conclusion beforehand. Therefore, I see this more as an application “white paper” rather than a primary research report.

Minor comments to figures and movies

Figure 3c: It would be beneficial to include a supplementary video of the 30-minute time-lapse showing the motility of individual microglia at 850 μm depth

Reply: Corresponding to Figure 3c we provide Supplementary Video 9 now, where the microglial motility recorded on two consecutive days is presented side by side.

Supp. Fig. 3: Imaging of the spinal cord dorsal horn gray matter at $\sim 400 \mu\text{m}$ depth. This might warrant some discussion, as there is a preprint by the Schaffer lab reporting deeper imaging ($>500 \mu\text{m}$; doi: 10.1101/2024.04.04.588110), albeit this might depend on the criteria of when an image is considered “useful”.

Reply: We thank the reviewer for the helpful comment. The preprint of the Schaffer lab reports blood vessel imaging up to a depth of $550 \mu\text{m}$. Their imaging of YFP-positive neurons in YFP-H tg mice was possible up to a depth of $400 \mu\text{m}$, which is comparable with our imaging depth ($380 \mu\text{m}$). We include the preprint from the Schaffer lab in our manuscript: “Thus, consistent with prior reports⁴⁸, it was not possible to achieve the depths of tissue penetration in spinal cord that we observed in other CNS regions. However, our system allowed for detailed view of neuronal architecture in spinal cord grey matter.” [p6 first paragraph]

Supp Fig. 5: What is the imaging depth for the Kenyon cells shown in Supp Fig. 5e?

Reply: The imaging depth of Kenyon cells in Suppl. Fig. 6e (previous Suppl. Fig. 5e) is 16 μm below the cuticle. The value was included in the legend.

Supp Fig. 4c: The YFP/GFP look-up table makes it difficult to distinguish overlapping microglia and neurons. What are the red signals in the three-photon panel?

Reply: We thank the reviewer for the helpful comment. We agree that it is difficult to distinguish yellow/green in Suppl. Fig. 5c (previous Fig. 4c). The images were generated by overlaying a green (GFP-channel) with a red (YFP-channel). Since YFP-neurons will generate fluorescence in both channels (green/red), they appear as yellow. Microglia are only present in the green GFP-Channel and appear green. Since there are also solely red structures fluorescent in the red channel, they appear red. We also don't exactly know what the red dots in Suppl. Fig. 5c are. We hypothesize that they could be autofluorescence generated by e.g. lipofuscin. To support our findings, we included an additional Supplementary Fig. 3 of a GFP-M::Cx3cr1-CreERT2::Rosa26tdTomato mouse to illustrate that microglia can be visualized in dorsal CA1 of the hippocampus.

Suppl. Fig. 3

Supp Video 5: The images at 750 μm and 1000 μm depth shown in Figure 2C don't align with the corresponding images in the video.

Reply: We thank the reviewer for careful reading. We corrected the values in Figure 2C, they are now corresponding to the video (now Suppl. Video 6).

Supp Video 9: The background is grainy with high noise. It would be helpful to visualize astrocyte structures by overlaying the cytoplasmic tdTom expression with GCaMP5g signals.

Reply: We thank the reviewer for the helpful comment. We included the td-Tomato channel and an overlay channel in the revised Suppl. Video 11 (previous Video 9).

Reviewer #2 (Remarks to the Author):

In this work, Fuhrmann et al. presented a new demonstration of applying three-photon (3P) in vivo imaging to achieve sub-cellular resolution in the medial prefrontal cortex (mPFC) of mice at unprecedented depths (up to 1700 μm). The mPFC is an important brain region for studying higher cognitive functions and has been of significant interest in neuroscience research. This is the first reported in vivo 3P imaging demonstration to achieve such depth in this area. This work demonstrated the imaging of both neuronal and glial activity, including dendritic spines, astrocytic Ca^{2+} -activity, and microglial processes, over extended periods with minimal invasiveness. The research highlights the advantages of 3P imaging over traditional two-photon (2P) methods, particularly in terms of depth penetration and imaging quality in deep brain structures.

Overall, I am very supportive of this work being published. The presented data is of high quality, and the demonstrations are generally convincing, with broad applications to various potential neuroscience problems. However, I have some technical comments that the authors need to address before final publication.

Reply: We very much thank the reviewer for the constructive and positive feedback.

Major comments:

1. My biggest concern is to achieve deeper imaging in the work, the authors chose a lateral imaging position to avoid imaging through highly myelinated corpus callosum layers which typically start to appear within 1.5 mm if imaging through somatocortex and deeper region as demonstrated in many other 3P mouse brain imaging work.

As mentioned in the paper "Since it is difficult to penetrate the highly myelinated corpus callosum in adult mice (4 months old), we chose a lateral imaging position to circumvent it (Supplementary Fig. 2a)." It would be better to discuss

a. How this location is searched and identified? This could be helpful protocol for other people interested in imaging deeper without the concern of white matter.

Reply: We thank the reviewer for the helpful comment. We now provide mediolateral coordinates (2.5 to 3.0 mm) and added boxes to the overview images in all figures concerning the hippocampus to better illustrate the lateral hippocampal imaging position.

b. Since it's less common to see 3P mPFC imaging results, it will be valuable to also learn the typical 3P imaging depth through white matter?

Reply: We thank the reviewer for the helpful comment. We include these values in the manuscript: "This enabled us to image hippocampal CA1 neurons deeper than 1 mm. In contrast, with 2P-imaging it was not possible to record YFP-expressing neurons deeper than 700-800 μm (**Supplementary Fig. 2a-c**). In younger mice (2.5 mo) 3P-imaging enabled imaging up to 1.3 mm with subcellular resolution of dendritic spines in dorsal CA1 (**Supplementary Fig. 3**).

c. What if a specific biological question requires not avoiding the highly myelinated corpus callosum, can the authors also show how deep 3P imaging can reach in their case, when imaging mPFC, while imaging depth of 1.7 mm without going through mPFC is impressive in value, imaging depth of the case going through mPFC is also valuable to report.

Reply: We thank the reviewer for the helpful comment. We included another Supplementary figure showing imaging in the hippocampus through the corpus callosum (new Supplementary Fig. 3). Here we were able to image up to 1.3 mm with subcellular resolution (dendritic spines) in young mice (2.5 mo).

Suppl. Fig. 3

2. The authors reported the spine density at various depth. This is impressive since it was previously reported this value can fluctuate due to animal motion. Particularly in this case, the demonstrated imaging was performed with head-fixed, awake, treadmill running mouse. What's the stability of the spine imaging, does this impact the reported values? For example, there could be some spines be out of focus or becoming more visible over a few minutes to hours imaging window due to animal motion. Can the authors comment on this?

Reply: Indeed, the Ca^{2+} -imaging in mPFC was recorded in awake mice. However, dendritic spines were recorded in isoflurane anesthesia as described in the methods. Nevertheless, awake recordings might be also possible for sub-cellular resolution. To correct for motion, we registered the images after recording. In addition, one could consider temporal and z-plane oversampling (taking an image of the same z-plane several times and at different depth) to correct for motion induced artifacts. Indeed, we acquired z-stacks spanning 10-30 μm with 1

µm z-steps. This ensures that dendritic spines can be retrieved even in long-term recordings. To find the same spines over several days, the position of neurons and their corresponding dendrites are recorded in an overview image. The dendritic branch pattern stays constant enabling us to retrieve the same dendritic fragment and spines over long periods of time. This method has been applied in many of our previous publications (Fuhrmann 2009, Gu 2014, Schmid 2015, Pfeiffer 2018)

3. My other big concern is related to this claim “we did not use any adaptive optics to correct for wavefront distortions” and related ones “We would like to stress that no adaptive optics for wavefront shaping were necessary to resolve these tiny structures more than a millimeter deep.” And “Our approach shows that subcellular resolution of dendritic spines, more than a millimeter deep, is achievable without adaptive optics”.

I highly doubt that this was due to the fact that the authors don't have to deal with highly myelinated corpus callosum which is the major challenge of high-resolution imaging of dendritic spines at depth and the need for adaptive optics there. The authors need to fairly compare with similar cases in other adaptive optics work, where many of them have gone through the white matter layer.

I suggest editing this sentence and other related discussion with adaptive optics to make a fairer comparison and claims. If the authors still manage to visualize sharp and clear spine imaging through white matter (highly myelinated corpus callosum), then this data should be provided and support this claim.

Reply: We thank the reviewer for the useful comment. Based on your suggestion, we imaged again through white matter in a 10 week old GFP-M::Cx3cr1-CreERT2::Rosa26tdTomato transgenic mouse (new Suppl. Fig. 3, see above). In young mice the CC is less developed than in older mice. Here, without the application of adaptive optics, we achieved subcellular resolution dendritic spine imaging in stratum oriens of the hippocampus. In older mice (Suppl. Fig. 2, 5) we were unable to resolve dendritic spines in the dorsal hippocampus through the corpus callosum. As suggested by the reviewer, we removed improper comparisons between HPC and mPFC with/without adaptive optics and refined our statement in the discussion concerning the HPC: “Adaptive optics have been used to improve the point spread function for sub-cellular resolution imaging in the hippocampus^{28,31,63} and might also be beneficial for spine resolution improvement in the mPFC. However, adaptive optics are another costly device in the beam-path that potentially broaden the pulse width, which needs proper correction and adjustment. Our approach shows that subcellular resolution of dendritic spines, more than a millimeter deep, is achievable without adaptive optics in the mPFC at any age and in the hippocampus of young mice.” [p9 last sentence until p10 first paragraph]

4. “Interestingly, although several studies have shown that the genetic composition of astrocytes differs strongly between superficial and deep cortical layers^{52,53}, our data indicate that astroglial Ca²⁺-activity, at least under anesthetized conditions, is uniform across all cortical layers.”

a. This sentence was not clear. What's the specific difference the authors refer to for astrocytes with different genetic composition?

Reply: This sentence was meant to make the point that we found a remarkably uniform level of baseline calcium activity across all cortical layers, whereas previous transcriptome analyses had shown layer-specific differences in gene expression. Hence, our data indicate that functional characteristics of astrocytes may differ from their genetic profile. We have now changed the text to make this point clearer: “Interestingly, although several studies have shown that gene expression profiles of astrocytes differs strongly between superficial and deep cortical layers^{53,54}, our data indicate that astroglial baseline Ca²⁺-activity – specifically amplitudes, temporal kinetics and activity frequencies – is uniform across all cortical layers

under anesthetized conditions, suggesting that functional characteristics of astrocytes may differ from their genetic profile.” [p10] (page second last paragraph).

b. What does it mean when the authors claim uniform activity across all cortical layers, is it about the distribution of certain activity-related parameters or what? Please clarify.

Reply: This was meant to interpret the data on amplitudes, temporal kinetics and activity frequencies of astroglial microdomains as well as somata, as reported in Fig 4. Specifically, we found that these three parameters of calcium activity, measured using 3P microscopy, were similar to amplitudes, temporal kinetics and activity frequencies in microdomains and somata of superficial cortical layers, indicating a “uniformity” of these parameters across layers. We have now changed the text to make this point clearer (see above).

5. Some comments on figure 1f and g in terms of SBR and power used:

a. Figure 1f, the initial power of 3P before 0.8 mm is lower than 2P, what’s the rationale there? Can the authors show the 2P and 3P power curve (signal versus power to validate a pure 2P and 3P response)?

Reply: We thank the reviewer for the helpful comment. The power of 3P in the first 800 μm is lower, because the minimum power of 3P-imaging throughout these depths was sufficient to achieve high SBR values and image quality. In addition, we tried to keep the 3P-power in the first 400 μm at minimum and then started to slowly increase to achieve good image quality. We also added a signal versus power curve in revised Fig. 1g.

b. Why the maximum power of 2P and 3P controlled to be the same 200mW? They have different wavelengths in use, so heating impact should be different. Thus, the maximum power control doesn’t have to be the same.

Reply: We thank the reviewer for the helpful comment. We agree that the heating at 200 mW avg power would be different for 2P and 3P and therefore the maximum power should not have been the same. However, we wanted to provide a user-friendly comparison, which will allow future users to apply 3P-imaging more easily. Therefore, we used it as a simplified comparison. In addition, the 200 mW 3P max power were only applied deep in the tissue. Here scattering will lead to a much smaller power in the focal volume. We refined the sentence in the following way: “We achieved imaging two times deeper with 3P-excitation compared to 2P-excitation, using a maximum laser power of 200 mW at maximum depths.” [page4, results first paragraph]

c. Why the initial SBR of 2P is higher than 3P? This needs explanation since 3P normally has higher SBR. Please refer:

- Figure 2b in Wang et al. “Quantitative analysis of 1300-nm three-photon calcium imaging in the mouse brain” eLife 2020
- Figure 3B in Takasaki et al. “Superficial bound of the depth limit of two-photon imaging in the mouse brain” eNeuro 2020
- Figure 2B in Akbari et al. “Imaging deeper than the transport mean free path with multiphoton microscopy” Biomed. Optics Express 2022

Reply: We thank the reviewer for the useful comment. Our explanation of the higher 2P versus 3P SBR within the first 400 μm may result from lower 3P excitation power. We kept 3P-excitation power at a minimum to avoid any phototoxicity especially up to a depth of 400 μm . To not confuse with previous findings, we changed our text and do not stress this difference anymore: “The signal to background ratio (SBR) of 2P-imaging rapidly decreased from 400 μm onwards to less than two at 500 – 600 μm depth. In contrast, the SBR of 3P-imaging only slightly decreased up to a depth of 1000 μm . Only at depths deeper than 1000 μm , the SBR started to decline (**Fig. 1f**).” [p4 end of first results paragraph]

6. All statistical analysis in figures need to complement with clear explanation in the caption on the meaning of the mid-line, error bars: are they mean, median, std? This applies to figure 2d, 3d, 3e, 3f, 4f, 5f-h and some supplementary figures.

Reply: We thank the reviewer for the useful comment. We applied the requested changes throughout the manuscript.

Reviewer #3 (Remarks to the Author):

In this manuscript, the authors demonstrated imaging of neurons and glial cells in the medial prefrontal cortex (mPFC) of mice using 1300-nm and 1650-nm three-photon microscopy. The authors established that three-photon imaging enables deep tissue imaging without invasive optical elements like GRIN lenses or microprisms, which are required by other optical imaging methods. The high variety and quality of experimental data make this work a valuable addition to the field. The authors adopted meticulous and thorough approaches towards experiments, and their data on optimizing optomechanical parameters for three-photon imaging provides valuable reference information for improving three-photon microscopy as a practical imaging modality. Overall, I strongly recommend its publication.

The authors demonstrated three-photon imaging of neurons and astrocytes up to 1.6 mm depth in the mouse mPFC, a region traditionally difficult to access optically. They rigorously compared three-photon to two-photon imaging depth, quantifying metrics such as signal-to-background ratio and required laser power. For chronic imaging, they impressively demonstrated the ability to resolve dendritic spines at depths beyond 1000 μm in mPFC. They also observed microglial process motility and studied calcium dynamics in astrocytes, finding uniform calcium activity patterns across cortical layers despite known genetic heterogeneity. The calcium imaging results in both mPFC neurons and dentate gyrus through a hippocampal window is also quite impressive. The technical details of the work seem quite solid to me, and I do not have any particular comment on them.

Reply: We very much thank the reviewer for the constructive and positive feedback.

One area where the manuscript could be improved is in comparing the practical tradeoffs between three-photon imaging and alternative techniques to image mPFC using microprisms or GRIN lenses. While the authors mention that their 4 x 3mm window offers a larger brain region to investigate compared to stably implanted GRIN lenses, a more detailed comparison of field-of-view sizes, imaging depths, and achievable resolution would be valuable. The authors note that microprism approaches are limited to superficial layers (1-3) of the mPFC, while their technique accesses deeper layers (3-5), but there is no fundamental reason why the microprism could not be inserted deeper. Also, these microprisms convert imaging breadth

to depth, and may lead to larger field-of-views. Therefore, the authors should have a more in-depth discussion on the comparison.

Reply: We thank the reviewer for the helpful comment. We agree with the reviewer that a more detailed comparison of field-of-view sizes, imaging depths and achievable resolution comparing GRIN, Prism and 3P approaches would be helpful. Therefore, we included a new Suppl. Fig. 7 to compare the approaches and revised the sentence in the text: “Ca²⁺-imaging in the mPFC of mice has been previously carried out through microprisms and GRIN-lenses³⁹⁻⁴¹. These approaches differ by the size of accessible brain region and invasiveness (Suppl. Fig. 7).”

In Suppl. Fig. 7, we focused on comparing imaging depth and field-of-view size, but excluded resolution comparisons for the different approaches. We did not include a resolution comparison, because resolution mainly depends the NA of the objective or GRIN lens and the wavelength, which usually have a wide range.

Concerning the L1-3 and L3-5 access, we referred to implantation of a prism into the fissure between the two hemispheres (Suppl. Fig. 7b), which would enable about 300 μ m penetration beyond the surface of the prism into prefrontal cortex. Thereby, only L1-3 neurons would be accessible, but not L3-5 neurons. Of course, one could also implant the microprism in 90° angle to midline enabling access to L1-5 in prefrontal cortex (Suppl. Fig. 7d). However, implanting the microprism in this orientation would require a cut into the brain and would be even more invasive than between the two hemispheres. We included these examples in the Suppl. Fig. 7. Concerning the size of prisms, we would like to note that prisms cannot easily be upscaled to e.g. 3x3 mm, because they would cause too much damage. In our hands, maximum prism size we have been implanting to access PFC or EC was 1.5 mm. This is significantly smaller in comparison to a 3 x 4 mm size image window that we used. Indeed, with larger cranial windows (e.g. 1 cm diameter) it would be possible to access even larger cortical areas with 3P-imaging.

Last but not least, the work would be more impactful if the authors choose to publicize their raw imaging data through a publicly accessible data repository. I would encourage the authors to do so, even though it is not mandatory.

Reply: We apologize that we did not upload the raw imaging data publicly. We have deposited the raw-data now at DRYAD (<http://datadryad.org/stash/share/NyFZ6zDehxF5eArX88Y12zvMn4cdEOgHiq8NOhOWzi8>).

Reply to Reviewers (revision 2):

Reviewer #2 (Remarks to the Author):

All my previous comments have been well addressed by the authors. I appreciate their efforts in conducting additional experiments with excellent results. I support its publication and believe it will be a valuable contribution to the three-photon brain imaging community.

Reply: Thank you very much for your efforts reviewing and improving our manuscript.

Reviewer #3 (Remarks to the Author):

I would like to thank the authors for addressing my comments, and I would like to reiterate my support on the publication of the manuscript.

Reply: Thank you very much for your support.

Reviewer #4 (Remarks to the Author):

The authors addressed my questions and comments.

Reply: Thank you very much.

Reply to Reviewers (revision 1):

Reviewers' comments:

Reviewer #1 (Remarks to the Author):

In this paper, Fuhrmann and colleagues demonstrate that three-photon imaging enables the imaging of several aspects of neural cell structure and dynamics in the deep-laying areas of the murine prefrontal cortex (with some images from the hippocampus and *Drosophila* brain). They describe in some detail their setup (which is semi-commercial) and some adjustments to the excitation laser pulses, e.g., via pulse compression. Several parameters are compared to two-photon imaging as the reference method, which, however, cannot penetrate many deeper-seated parts of the medial prefrontal cortex. The paper's novelty claim is that this represents the first application of three-photon microscopy to deep prefrontal cortex areas, which is of substantial interest in behavioral paradigms and disease models.

Overall, I found the paper technically sound and easy to read. My problem is that it contains hardly any surprising results – while it is nice to see that three-photon microscopy can reach the deep prefrontal cortex, I think this could be deduced from first principles and prior literature. Indeed, three-photon imaging has long been demonstrated to enable deeper imaging penetration of brain tissue in substantial prior work by other labs (see, e.g., Horton Nat Phot 2013; most recently, Thornton et al., Nat Neuro 2024, but also miniature 3P systems, Klioutchnikov Nat Meth 2020).

Thus, while I think the paper contains an interesting application of a well-documented technique, I would argue that this does not justify a stand-alone report – after all, “deep imaging” is the established rationale why three-photon microscopes or kits to built them are now sold by several commercial providers; hence any likely reader of this article would typically concur with its conclusion beforehand. Therefore, I see this more as an application “white paper” rather than a primary research report.

Minor comments to figures and movies

Figure 3c: It would be beneficial to include a supplementary video of the 30-minute time-lapse showing the motility of individual microglia at 850 μm depth

Reply: Corresponding to Figure 3c we provide Supplementary Video 9 now, where the microglial motility recorded on two consecutive days is presented side by side.

Supp. Fig. 3: Imaging of the spinal cord dorsal horn gray matter at $\sim 400 \mu\text{m}$ depth. This might warrant some discussion, as there is a preprint by the Schaffer lab reporting deeper imaging ($>500 \mu\text{m}$; doi: 10.1101/2024.04.04.588110), albeit this might depend on the criteria of when an image is considered “useful”.

Reply: We thank the reviewer for the helpful comment. The preprint of the Schaffer lab reports blood vessel imaging up to a depth of $550 \mu\text{m}$. Their imaging of YFP-positive neurons in YFP-H tg mice was possible up to a depth of $400 \mu\text{m}$, which is comparable with our imaging depth ($380 \mu\text{m}$). We include the preprint from the Schaffer lab in our manuscript: “Thus, consistent with prior reports⁴⁸, it was not possible to achieve the depths of tissue penetration in spinal cord that we observed in other CNS regions. However, our system allowed for detailed view of neuronal architecture in spinal cord grey matter.” [p6 first paragraph]

Supp Fig. 5: What is the imaging depth for the Kenyon cells shown in Supp Fig. 5e?

Reply: The imaging depth of Kenyon cells in Suppl. Fig. 6e (previous Suppl. Fig. 5e) is $16 \mu\text{m}$ below the cuticle. The value was included in the legend.

Supp Fig. 4c: The YFP/GFP look-up table makes it difficult to distinguish overlapping microglia and neurons. What are the red signals in the three-photon panel?

Reply: We thank the reviewer for the helpful comment. We agree that it is difficult to distinguish yellow/green in Suppl. Fig. 5c (previous Fig. 4c). The images were generated by overlaying a green (GFP-channel) with a red (YFP-channel). Since YFP-neurons will generate fluorescence in both channels (green/red), they appear as yellow. Microglia are only present in the green GFP-Channel and appear green. Since there are also solely red structures fluorescent in the red channel, they appear red. We also don't exactly know what the red dots in Suppl. Fig. 5c are. We hypothesize that they could be autofluorescence generated by e.g. lipofuscin. To support our findings, we included an additional Supplementary Fig. 3 of a GFP-M::Cx3cr1-CreERT2::Rosa26tdTomato mouse to illustrate that microglia can be visualized in dorsal CA1 of the hippocampus.

Suppl. Fig. 3

Supp Video 5: The images at 750 µm and 1000 µm depth shown in Figure 2C don't align with the corresponding images in the video.

Reply: We thank the reviewer for careful reading. We corrected the values in Figure 2C, they are now corresponding to the video (now Suppl. Video 6).

Supp Video 9: The background is grainy with high noise. It would be helpful to visualize astrocyte structures by overlaying the cytoplasmic tdTom expression with GCaMP5g signals.

Reply: We thank the reviewer for the helpful comment. We included the td-Tomato channel and an overlay channel in the revised Suppl. Video 11 (previous Video 9).

Reviewer #2 (Remarks to the Author):

In this work, Fuhrmann et al. presented a new demonstration of applying three-photon (3P) in vivo imaging to achieve sub-cellular resolution in the medial prefrontal cortex (mPFC) of mice at unprecedented depths (up to 1700 μm). The mPFC is an important brain region for studying higher cognitive functions and has been of significant interest in neuroscience research. This is the first reported in vivo 3P imaging demonstration to achieve such depth in this area. This work demonstrated the imaging of both neuronal and glial activity, including dendritic spines, astrocytic Ca²⁺-activity, and microglial processes, over extended periods with minimal invasiveness. The research highlights the advantages of 3P imaging over traditional two-photon (2P) methods, particularly in terms of depth penetration and imaging quality in deep brain structures.

Overall, I am very supportive of this work being published. The presented data is of high quality, and the demonstrations are generally convincing, with broad applications to various potential neuroscience problems. However, I have some technical comments that the authors need to address before final publication.

Reply: We very much thank the reviewer for the constructive and positive feedback.

Major comments:

1. My biggest concern is to achieve deeper imaging in the work, the authors chose a lateral imaging position to avoid imaging through highly myelinated corpus callosum layers which typically start to appear within 1.5 mm if imaging through somatocortex and deeper region as demonstrated in many other 3P mouse brain imaging work.

As mentioned in the paper “Since it is difficult to penetrate the highly myelinated corpus callosum in adult mice (4 months old), we chose a lateral imaging position to circumvent it (Supplementary Fig. 2a).” It would be better to discuss

a. How this location is searched and identified? This could be helpful protocol for other people interested in imaging deeper without the concern of white matter.

Reply: We thank the reviewer for the helpful comment. We now provide mediolateral coordinates (2.5 to 3.0 mm) and added boxes to the overview images in all figures concerning the hippocampus to better illustrate the lateral hippocampal imaging position.

b. Since it's less common to see 3P mPFC imaging results, it will be valuable to also learn the typical 3P imaging depth through white matter?

Reply: We thank the reviewer for the helpful comment. We include these values in the manuscript: "This enabled us to image hippocampal CA1 neurons deeper than 1 mm. In contrast, with 2P-imaging it was not possible to record YFP-expressing neurons deeper than 700-800 μm (**Supplementary Fig. 2a-c**). In younger mice (2.5 mo) 3P-imaging enabled imaging up to 1.3 mm with subcellular resolution of dendritic spines in dorsal CA1 (**Supplementary Fig. 3**).

c. What if a specific biological question requires not avoiding the highly myelinated corpus callosum, can the authors also show how deep 3P imaging can reach in their case, when imaging mPFC, while imaging depth of 1.7 mm without going through mPFC is impressive in value, imaging depth of the case going through mPFC is also valuable to report.

Reply: We thank the reviewer for the helpful comment. We included another Supplementary figure showing imaging in the hippocampus through the corpus callosum (new Supplementary Fig. 3). Here we were able to image up to 1.3 mm with subcellular resolution (dendritic spines) in young mice (2.5 mo).

Suppl. Fig. 3

2. The authors reported the spine density at various depth. This is impressive since it was previously reported this value can fluctuate due to animal motion. Particularly in this case, the demonstrated imaging was performed with head-fixed, awake, treadmill running mouse. What's the stability of the spine imaging, does this impact the reported values? For example, there could be some spines be out of focus or becoming more visible over a few minutes to hours imaging window due to animal motion. Can the authors comment on this?

Reply: Indeed, the Ca^{2+} -imaging in mPFC was recorded in awake mice. However, dendritic spines were recorded in isoflurane anesthesia as described in the methods. Nevertheless, awake recordings might be also possible for sub-cellular resolution. To correct for motion, we registered the images after recording. In addition, one could consider temporal and z-plane oversampling (taking an image of the same z-plane several times and at different depth) to correct for motion induced artifacts. Indeed, we acquired z-stacks spanning 10-30 µm with 1

µm z-steps. This ensures that dendritic spines can be retrieved even in long-term recordings. To find the same spines over several days, the position of neurons and their corresponding dendrites are recorded in an overview image. The dendritic branch pattern stays constant enabling us to retrieve the same dendritic fragment and spines over long periods of time. This method has been applied in many of our previous publications (Fuhrmann 2009, Gu 2014, Schmid 2015, Pfeiffer 2018)

3. My other big concern is related to this claim “we did not use any adaptive optics to correct for wavefront distortions” and related ones “We would like to stress that no adaptive optics for wavefront shaping were necessary to resolve these tiny structures more than a millimeter deep.” And “Our approach shows that subcellular resolution of dendritic spines, more than a millimeter deep, is achievable without adaptive optics”.

I highly doubt that this was due to the fact that the authors don't have to deal with highly myelinated corpus callosum which is the major challenge of high-resolution imaging of dendritic spines at depth and the need for adaptive optics there. The authors need to fairly compare with similar cases in other adaptive optics work, where many of them have gone through the white matter layer.

I suggest editing this sentence and other related discussion with adaptive optics to make a fairer comparison and claims. If the authors still manage to visualize sharp and clear spine imaging through white matter (highly myelinated corpus callosum), then this data should be provided and support this claim.

Reply: We thank the reviewer for the useful comment. Based on your suggestion, we imaged again through white matter in a 10 week old GFP-M::Cx3cr1-CreERT2::Rosa26tdTomato transgenic mouse (new Suppl. Fig. 3, see above). In young mice the CC is less developed than in older mice. Here, without the application of adaptive optics, we achieved subcellular resolution dendritic spine imaging in stratum oriens of the hippocampus. In older mice (Suppl. Fig. 2, 5) we were unable to resolve dendritic spines in the dorsal hippocampus through the corpus callosum. As suggested by the reviewer, we removed improper comparisons between HPC and mPFC with/without adaptive optics and refined our statement in the discussion concerning the HPC: “Adaptive optics have been used to improve the point spread function for sub-cellular resolution imaging in the hippocampus^{28,31,63} and might also be beneficial for spine resolution improvement in the mPFC. However, adaptive optics are another costly device in the beam-path that potentially broaden the pulse width, which needs proper correction and adjustment. Our approach shows that subcellular resolution of dendritic spines, more than a millimeter deep, is achievable without adaptive optics in the mPFC at any age and in the hippocampus of young mice.” [p9 last sentence until p10 first paragraph]

4. “Interestingly, although several studies have shown that the genetic composition of astrocytes differs strongly between superficial and deep cortical layers^{52,53}, our data indicate that astroglial Ca²⁺-activity, at least under anesthetized conditions, is uniform across all cortical layers.”

a. This sentence was not clear. What's the specific difference the authors refer to for astrocytes with different genetic composition?

Reply: This sentence was meant to make the point that we found a remarkably uniform level of baseline calcium activity across all cortical layers, whereas previous transcriptome analyses had shown layer-specific differences in gene expression. Hence, our data indicate that functional characteristics of astrocytes may differ from their genetic profile. We have now changed the text to make this point clearer: “Interestingly, although several studies have shown that gene expression profiles of astrocytes differs strongly between superficial and deep cortical layers^{53,54}, our data indicate that astroglial baseline Ca²⁺-activity – specifically amplitudes, temporal kinetics and activity frequencies – is uniform across all cortical layers

under anesthetized conditions, suggesting that functional characteristics of astrocytes may differ from their genetic profile.” [p10] (page second last paragraph).

b. What does it mean when the authors claim uniform activity across all cortical layers, is it about the distribution of certain activity-related parameters or what? Please clarify.

Reply: This was meant to interpret the data on amplitudes, temporal kinetics and activity frequencies of astroglial microdomains as well as somata, as reported in Fig 4. Specifically, we found that these three parameters of calcium activity, measured using 3P microscopy, were similar to amplitudes, temporal kinetics and activity frequencies in microdomains and somata of superficial cortical layers, indicating a “uniformity” of these parameters across layers. We have now changed the text to make this point clearer (see above).

5. Some comments on figure 1f and g in terms of SBR and power used:

a. Figure 1f, the initial power of 3P before 0.8 mm is lower than 2P, what’s the rationale there? Can the authors show the 2P and 3P power curve (signal versus power to validate a pure 2P and 3P response)?

Reply: We thank the reviewer for the helpful comment. The power of 3P in the first 800 μm is lower, because the minimum power of 3P-imaging throughout these depths was sufficient to achieve high SBR values and image quality. In addition, we tried to keep the 3P-power in the first 400 μm at minimum and then started to slowly increase to achieve good image quality. We also added a signal versus power curve in revised Fig. 1g.

b. Why the maximum power of 2P and 3P controlled to be the same 200mW? They have different wavelengths in use, so heating impact should be different. Thus, the maximum power control doesn’t have to be the same.

Reply: We thank the reviewer for the helpful comment. We agree that the heating at 200 mW avg power would be different for 2P and 3P and therefore the maximum power should not have been the same. However, we wanted to provide a user-friendly comparison, which will allow future users to apply 3P-imaging more easily. Therefore, we used it as a simplified comparison. In addition, the 200 mW 3P max power were only applied deep in the tissue. Here scattering will lead to a much smaller power in the focal volume. We refined the sentence in the following way: “We achieved imaging two times deeper with 3P-excitation compared to 2P-excitation, using a maximum laser power of 200 mW at maximum depths.” [page4, results first paragraph]

c. Why the initial SBR of 2P is higher than 3P? This needs explanation since 3P normally has higher SBR. Please refer:

- Figure 2b in Wang et al. “Quantitative analysis of 1300-nm three-photon calcium imaging in the mouse brain” eLife 2020
- Figure 3B in Takasaki et al. “Superficial bound of the depth limit of two-photon imaging in the mouse brain” eNeuro 2020
- Figure 2B in Akbari et al. “Imaging deeper than the transport mean free path with multiphoton microscopy” Biomed. Optics Express 2022

Reply: We thank the reviewer for the useful comment. Our explanation of the higher 2P versus 3P SBR within the first 400 μm may result from lower 3P excitation power. We kept 3P-excitation power at a minimum to avoid any phototoxicity especially up to a depth of 400 μm . To not confuse with previous findings, we changed our text and do not stress this difference anymore: “The signal to background ratio (SBR) of 2P-imaging rapidly decreased from 400 μm onwards to less than two at 500 – 600 μm depth. In contrast, the SBR of 3P-imaging only slightly decreased up to a depth of 1000 μm . Only at depths deeper than 1000 μm , the SBR started to decline (**Fig. 1f**).” [p4 end of first results paragraph]

6. All statistical analysis in figures need to complement with clear explanation in the caption on the meaning of the mid-line, error bars: are they mean, median, std? This applies to figure 2d, 3d, 3e, 3f, 4f, 5f-h and some supplementary figures.

Reply: We thank the reviewer for the useful comment. We applied the requested changes throughout the manuscript.

Reviewer #3 (Remarks to the Author):

In this manuscript, the authors demonstrated imaging of neurons and glial cells in the medial prefrontal cortex (mPFC) of mice using 1300-nm and 1650-nm three-photon microscopy. The authors established that three-photon imaging enables deep tissue imaging without invasive optical elements like GRIN lenses or microprisms, which are required by other optical imaging methods. The high variety and quality of experimental data make this work a valuable addition to the field. The authors adopted meticulous and thorough approaches towards experiments, and their data on optimizing optomechanical parameters for three-photon imaging provides valuable reference information for improving three-photon microscopy as a practical imaging modality. Overall, I strongly recommend its publication.

The authors demonstrated three-photon imaging of neurons and astrocytes up to 1.6 mm depth in the mouse mPFC, a region traditionally difficult to access optically. They rigorously compared three-photon to two-photon imaging depth, quantifying metrics such as signal-to-background ratio and required laser power. For chronic imaging, they impressively demonstrated the ability to resolve dendritic spines at depths beyond 1000 μm in mPFC. They also observed microglial process motility and studied calcium dynamics in astrocytes, finding uniform calcium activity patterns across cortical layers despite known genetic heterogeneity. The calcium imaging results in both mPFC neurons and dentate gyrus through a hippocampal window is also quite impressive. The technical details of the work seem quite solid to me, and I do not have any particular comment on them.

Reply: We very much thank the reviewer for the constructive and positive feedback.

One area where the manuscript could be improved is in comparing the practical tradeoffs between three-photon imaging and alternative techniques to image mPFC using microprisms or GRIN lenses. While the authors mention that their 4 x 3mm window offers a larger brain region to investigate compared to stably implanted GRIN lenses, a more detailed comparison of field-of-view sizes, imaging depths, and achievable resolution would be valuable. The authors note that microprism approaches are limited to superficial layers (1-3) of the mPFC, while their technique accesses deeper layers (3-5), but there is no fundamental reason why the microprism could not be inserted deeper. Also, these microprisms convert imaging breadth

to depth, and may lead to larger field-of-views. Therefore, the authors should have a more in-depth discussion on the comparison.

Reply: We thank the reviewer for the helpful comment. We agree with the reviewer that a more detailed comparison of field-of-view sizes, imaging depths and achievable resolution comparing GRIN, Prism and 3P approaches would be helpful. Therefore, we included a new Suppl. Fig. 7 to compare the approaches and revised the sentence in the text: “Ca²⁺-imaging in the mPFC of mice has been previously carried out through microprisms and GRIN-lenses³⁹⁻⁴¹. These approaches differ by the size of accessible brain region and invasiveness (Suppl. Fig. 7).”

In Suppl. Fig. 7, we focused on comparing imaging depth and field-of-view size, but excluded resolution comparisons for the different approaches. We did not include a resolution comparison, because resolution mainly depends the NA of the objective or GRIN lens and the wavelength, which usually have a wide range.

Concerning the L1-3 and L3-5 access, we referred to implantation of a prism into the fissure between the two hemispheres (Suppl. Fig. 7b), which would enable about 300 μ m penetration beyond the surface of the prism into prefrontal cortex. Thereby, only L1-3 neurons would be accessible, but not L3-5 neurons. Of course, one could also implant the microprism in 90° angle to midline enabling access to L1-5 in prefrontal cortex (Suppl. Fig. 7d). However, implanting the microprism in this orientation would require a cut into the brain and would be even more invasive than between the two hemispheres. We included these examples in the Suppl. Fig. 7. Concerning the size of prisms, we would like to note that prisms cannot easily be upscaled to e.g. 3x3 mm, because they would cause too much damage. In our hands, maximum prism size we have been implanting to access PFC or EC was 1.5 mm. This is significantly smaller in comparison to a 3 x 4 mm size image window that we used. Indeed, with larger cranial windows (e.g. 1 cm diameter) it would be possible to access even larger cortical areas with 3P-imaging.

Last but not least, the work would be more impactful if the authors choose to publicize their raw imaging data through a publicly accessible data repository. I would encourage the authors to do so, even though it is not mandatory.

Reply: We apologize that we did not upload the raw imaging data publicly. We have deposited the raw-data now at DRYAD (<http://datadryad.org/stash/share/NyFZ6zDehxF5eArX88Y12zvMn4cdEOgHiq8NOhOWzi8>).